
# Stieltjes Functions and Spectral Analysis
# in the Physics of Sea Ice

Kenneth M. Golden[1], N. Benjamin Murphy[1], and Elena Cherkaev[1]

[1]Department of Mathematics, University of Utah , 155 S 1400 E RM 233, Salt Lake City, UT 84112-0090

**Correspondence:** Kenneth M. Golden (golden@math.utah.edu)

**Abstract.** Polar sea ice is a critical component of Earth's climate system. As a material it is a multiscale composite with temperature dependent millimeter-scale brine microstructure, and centimeter-scale polycrystalline microstructure which is largely determined by how the ice was formed. The surface layer of the polar oceans can be viewed as a granular composite of ice floes in a sea water host, with floe sizes ranging from centimeters to tens of kilometers. A principal challenge in modeling sea ice and its role in climate is how to use information on smaller scale structure to find the effective or homogenized properties on larger scales relevant to process studies and coarse-grained climate models. That is, how do you predict macroscopic behavior from microscopic laws, like in statistical mechanics and solid state physics? Also of great interest in climate science is the inverse problem of recovering parameters controlling small scale processes from large scale observations. Motivated by sea ice remote sensing, the analytic continuation method for obtaining rigorous bounds on the homogenized coefficients of two phase composites was applied to the complex permittivity of sea ice, which is a Stieltjes function of the ratio of the permittivities of ice and brine. Integral representations for the effective parameters distill the complexities of the composite microgeometry into the spectral properties of a self-adjoint operator like the Hamiltonian in quantum physics. These techniques have been extended to polycrystalline materials, advection diffusion processes, and ocean waves in the sea ice cover. Here we discuss this powerful approach in homogenization, highlighting the spectral representations and resolvent structure of the fields that are shared by the two component theory and its extensions. Spectral analysis of sea ice structures leads to a random matrix theory picture of percolation processes in composites, establishing parallels to Anderson localization and semiconductor physics, which then provides new insights into the physics of sea ice.

## 1 Introduction

The precipitous loss of nearly half the extent of the summer Arctic sea ice cover over the past four decades or so, since satellite observations started in 1979, is perhaps one of the most visible large-scale changes on Earth's surface connected to planetary warming, with significant implications for the Arctic and beyond Stroeve et al. (2007, 2012); Maslanik et al. (2007); Notz and Community (2020); Notz and Stroeve (2016). While the response of the sea ice pack surrounding the Antarctic continent to the changing climate has perhaps not been as clear as in the Arctic, this past year the summer sea ice extent set a record low Turner et al. (2022). The emerging dynamics of Earth's polar marine environments are complex and highly variable. Yet they are increasingly important to understand and predict, as the sea ice packs form a key component of the climate system,


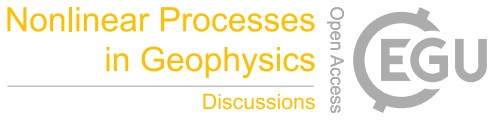

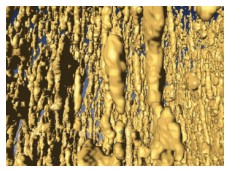 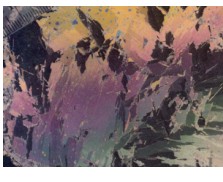 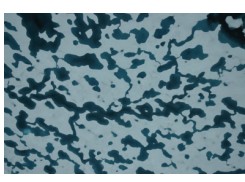 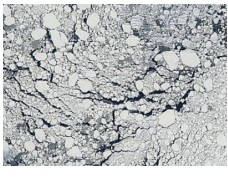 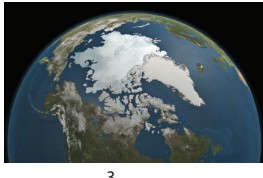

| millimeters | centimeters | meters | kilometers | $10^3$ kilometers |

**Figure 1. Sea ice as a multiscale composite material.** From left to right: millimeter-scale brine inclusions that form the porous microstructure of sea ice Golden et al. (2007); centimeter-scale polycrystalline structure of sea ice Arcone et al. (1986); melt ponds on Arctic sea ice in late spring and summer (D. Perovich) turn the surface into a two phase composite of ice and melt water; the sea ice pack as a granular composite viewed from space (NASA), with "grains" ranging in horizontal extent from meters to tens of kilometers; the Arctic Ocean viewed from space (NASA).

are indicators of our changing climate, and directly impact expanding human activities in these regions. Sea ice has bearing on almost any study of the physics or biology of the polar marine system, as well as on almost any maritime operations or logistics. Advancing our ability to analyze, model, and predict the behavior of sea ice is critical to improving projections of climate change and the response of polar ecosystems, and in meeting the challenges of increased human activities in the Arctic
Golden et al. (2020).

One of the fascinating, yet challenging aspects of modeling sea ice and its role in global climate is the sheer range of relevant length scales − over ten orders of magnitude, from the sub-millimeter scale to thousands of kilometers, as indicated in Figure 1. Modeling the macroscopic behavior of sea ice on scales appropriate for climate models or for process studies depends on understanding the properties of sea ice on finer scales, down to individual floes and even the scale of the brine inclusions which
control so many of the distinct physical characteristics of sea ice as a material. Climate models challenge the most powerful supercomputers to their fullest capacity. However, even the largest computers still limit the resolution to tens of kilometers and typically require clever approximations and parameterizations to incorporate the basic physics of sea ice Golden et al. (2020); Golden (2015, 2009). One of the fundamental challenges in modeling sea ice—and a central theme in what follows—is how to account for the influence of the microscale on macroscopic behavior, that is, how to rigorously use information about smaller
scales to predict effective behavior on larger scales. Here we consider three different homogenization problems in the physics of sea ice: the classic two phase problem of brine inclusions in an ice host, sea ice as a polycrystalline material, and advection diffusion processes such as thermal conduction or nutrient diffusion in the presence of, e.g. convective brine flow. All of these questions are also of particular interest in polar microbial ecology Thomas and Dieckmann (2003); Reimer et al. (2022).

We observe that this central problem of studying the effective properties of sea ice is analogous to the main focus of statistical
mechanics where knowledge of molecular interactions or microscopic laws is used to find collective or macroscopic behavior Thompson (1988); Christensen and Moloney (2005). Moreover, it also shares fundamental similarities with homogenization theory for composites where larger scale effective properties are calculated from knowledge of the microstructure Milton (2002); Torquato (2002); Bensoussan et al. (1978); Papanicolaou and Varadhan (1982); Kozlov (1989). These fields of physics

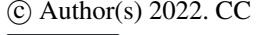



and applied mathematics provide a natural framework for treating sea ice in predictive models of climate, and improving
projections of how Earth's polar ice packs may evolve in the future.

The analytic continuation method (ACM) Bergman (1980); Milton (1980); Golden and Papanicolaou (1983); Golden (1997b); Milton (2002) in particular, yields powerful integral representations for the effective or homogenized transport coefficients of two component Golden and Papanicolaou (1983) or multicomponent Golden and Papanicolaou (1985); Golden (1986) media. The method exploits the properties of these coefficients as analytic functions of ratios of the constituent parameters for two
phase media, such as the ratio of the electrical or thermal conductivities, or the complex permittivities. The geometry of the composite microstructure is encoded into a self-adjoint operator $G$ through the characteristic function which takes the values 1 in one component (brine) and 0 in the other (ice). The key step in obtaining the integral representation, say in the case of electrical conductivity, is to derive a formula for the local electric field in terms of the resolvent of $G$, and then apply the spectral theorem in an appropriate Hilbert space. This representation for the effective conductivity (or effective complex permittivity)
achieves a complete separation between the component parameters in the variable, and the geometry of the microstructure embedded in the *spectral measure* of $G$, the principal mathematical object in the integral. In a discrete model of a composite, the operator $G$ becomes a random matrix, whose eigenvalues and eigenvectors can be used to compute the spectral measure Murphy et al. (2015).

The Stieltjes or Herglotz structure of the effective parameters and their integral representations can be exploited to use the
moments of the spectral measure, or the correlation functions of the composite microstructure, to find rigorous bounds on the homogenized transport coefficients Bergman (1980); Milton (1980); Golden and Papanicolaou (1983); Golden (1986); Baker and Graves-Morris (1996); Milton (2002). Bounds on the complex permittivity of sea ice as a two phase composite were first obtained in the context of remote sensing and the mathematical analysis of sea ice electromagnetic properties Golden (1995); Golden et al. (1998c, b). For example, the mass of the spectral measure is the brine volume fraction. If this is known, then one
can obtain *elementary bounds* in the complex case, which reduce to the classical arithmetic and harmonic mean bounds for real parameters. If the material is further assumed to be statistically isotropic, then tighter Hashin-Shtrikman bounds can be obtained. Even tighter bounds can be obtained when the composite is assumed to have *matrix-particle* structure, such as separated brine inclusions in a pure ice host Bruno (1991); Golden (1997b), which leads to gaps in the spectrum of $G$, and tighter constraints on the support of the spectral measure. In remote sensing the inverse homogenization problem, Cherkaev and Golden
(1998); Cherkaev (2001), where knowledge of bulk electromagnetic behavior, such as measurements of the effective complex permittivity, is inverted to obtain the spectral measure Cherkaev (2001) or bounds on the microstructural characteristics such as the brine volume fraction Cherkaev and Golden (1998); Golden et al. (1998b); Gully et al. (2007); Cherkaev and Bonifasi-Lista (2011), crystal orientation Gully et al. (2015), and connectivity Orum et al. (2012). The *microscale* structure, which determines the spectral measure and the homogenized coefficient, is thus linked to the *macroscopic* behavior via the operator $G$ and its
spectral characteristics, and *vice versa*. In the multicomponent case with three or more constituents, the homogenized transport coefficients are analytic functions of two or more complex variables, and a polydisc representation formula was exploited to obtain bounds Golden and Papanicolaou (1985); Golden (1986).





The first area of application where the ACM was extended beyond the classical case of two component and multiphase composites is diffusive transport in the presence of a flow field, which is widely encountered throughout science and engineering
McLaughlin et al. (1985); Biferale et al. (1995); Fannjiang and Papanicolaou (1994, 1997); Pavliotis (2002); Majda and Kramer (1999); Majda and Souganidis (1994); Xin (2009). In addition to thermal, saline, and nutrient transport through the porous microstructure of sea ice, large scale transport of ice floes and heat are also advection diffusion processes. Avellaneda and Majda Avellaneda and Majda (1989, 1991) found a Stieltjes integral representation for the effective diffusivity as a function of the Péclet number for diffusion in an incompressible velocity field. Based on the approach in Golden and Papanicolaou (1983),
they set up a Hilbert space framework and applied the spectral theorem to a resolvent representation involving analogues of $G$ and the electric field, where the spectral measure depends on the geometry of the velocity field, and knowledge of its moments yields bounds on the effective diffusivity. In Murphy et al. (2017b, 2020) we proved novel versions of the Stieltjes formulas, developed a framework to numerically compute the spectral measures and a systematic method to find its moments − and thus a hierarchy of bounds, for both the time dependent and independent cases.

In another extension of the ACM to a large class of media, a Stieltjes integral representation and rigorous bounds for the effective complex permittivity of polycrystalline media were developed in Gully et al. (2015), based on a resolvent formula for the electric field, and earlier observations in Milton (1981); Bergman and Stroud (1992); Milton (2002). The bounds assume knowledge of the average crystal orientation and the complex permittivity tensor of an individual crystal grain. In sea ice, finding the complex permittivity tensor of an individual crystal involves homogenizing the smaller scale brine microstructure Gully
et al. (2015). The polycrystalline structure of sea ice, as characterized by the statistics of grain size, shape, and orientation, is influenced by the conditions under which the ice was grown Weeks and Ackley (1982); Petrich and Eicken (2009); Untersteiner (1986). For example, while sea ice grown in quiescent conditions tends to have rather large-grained *columnar* structure, when grown in more turbulent or wavy conditions it typically has a fine-grained *granular* structure. These distinctly different ice types have quite different fluid flow properties Golden et al. (1998a, 2022). Also, when there is a well-defined current direction
during formation, crystal orientations tend to be statistically anisotropic within the horizontal plane Weeks and Gow (1980), which can significantly affect the sea ice radar signature, and measurements of sea ice thickness and properties used to validate climate models Golden and Ackley (1981); McLean et al. (2022).

The interaction of ocean surface waves with polar sea ice is a critical process in Earth's climate system; its accurate representation is of great importance for developing efficient climate models. Ice-ocean interactions have become increasingly
important in the Arctic with the precipitous declines of summer sea ice extent and increases in wave activity Waseda et al. (2018), while at the same time the marginal ice zone (MIZ), which is characterized by strong wave-ice and atmosphere-ice-ocean interactions, has widened significantly Strong and Rigor (2013). These recent changes can have complex implications for both sea ice formation and melting Li et al. (2021). Indeed, the propagation of surface waves through Earth's sea ice covers is a complex phenomenon that drives their growth and decay. One of the main approaches to studying waves in sea ice which is
valid when wavelengths are much greater than floe sizes, is to model the surface layer of the ice-covered ocean as a continuum with effective properties Bates and Shapiro (1980); Keller (1998); Wang and Shen (2010); Mosig et al. (2015). Recently this fundamental problem in sea ice physics was homogenized, with a Stieltjes representation for the effective complex viscoelas-





ticity of the surface layer, based on a resolvent formula for the local strain field. The integral involves a spectral measure of a self-adjoint operator depending on the geometry of the floe configurations. If its mass, or ice concentration, is known then

rigorous bounds on the complex viscoelasticity are obtained in Sampson et al. (2022). Previously this effective parameter had only been fitted to wave data. We will leave any detailed discussion of waves in sea ice to other publications.

Early on in our work in extending the ACM to the above problems in sea ice physics, it was clear that the classical approach based on bounding effective parameters using the moments of the spectral measure would in many cases have limited effectiveness. Bounds with only a moment or two known can be quite wide, particularly for a high contrast in the properties of the

constituents, like in sea ice. We then developed a framework in the classic two phase case for computing the spectral measure through discretization of the relevant microstructures and finding the eigenvalues and eigenvectors of the matrix representation of $G$. By developing the mathematical foundation for these computations Murphy et al. (2015) and studying the properties of computed spectral measures for a broad range of sea ice and other microstructures, like human bone Golden et al. (2011), we discovered that the statistics of the eigenvalues displayed fascinating behavior depending on the connectedness of one of the

phases.

The statistical behavior of the spectrum is related to the extent that the eigenfunctions overlap. A key example is the Anderson theory of the metal-insulator transition (MIT) Anderson (1958); Evers and Mirlin (2008), which provides a powerful theoretical framework for understanding when a disordered medium allows electronic transport, and when it does not. Indeed, for large enough disorder the electrons are localized in different places, with uncorrelated energy levels described by Poisson statistics

Shklovskii et al. (1993); Kravtsov and Muttalib (1997). For small disorder, the wave functions are extended and overlap, giving rise to correlated Wigner-Dyson (WD) statistics Shklovskii et al. (1993); Kravtsov and Muttalib (1997) with strong level repulsion Guhr et al. (1998). In work on the effective complex permittivity for electromagnetic wave propagation through two phase composites in the long wavelength regime (or any other transport coefficient like thermal or electrical conductivity), we found an Anderson transition in spectral characteristics as the microstructure developed long range order in the approach

to a percolation threshold Murphy et al. (2017a). We observe transitions in localization characteristics of the field vectors and associated transitions in spectral behavior from uncorrelated Poissonian statistics to universal (repulsive) Wigner-Dyson statistics, connected to the Gaussian Orthogonal Ensemble (GOE) in random matrix theory. Mobility edges appear, analogous to Anderson localization where they mark the characteristic energies of the quantum MIT Guhr et al. (1998). In Morison et al. (2022) a novel class of two phase composites was introduced, based on Moiré patterns, that display exotic effective properties,

and dramatic transitions in spectral behavior with very small changes in system parameters.

Over the past decade or so we have laid the groundwork for significant advances in the mathematical modeling of sea ice processes by developing Stieltjes integral representations for homogenized parameters in several new contexts of importance in the physics of sea ice and its role in climate. We focus on the central role that the spectral measure plays in determining effective behavior. The analytic continuation method is a powerful approach in homogenization that provides a robust mathe-

matical framework for rigorously studying effective properties in the sea ice system. The body of work that is discussed here will advance our sea ice modeling capabilities and how sea ice is represented in global climate models, which will improve projections of the fate of sea ice and the ecosystems it supports. Moreover, the functions we study here in the sea ice context





share the same mathematical properties as effective parameters in many other areas of science and engineering, so our work
will advance knowledge of these other materials as well, as evidenced for example by Morison et al. (2022), Golden et al.
(2011) and Gully et al. (2015).

## 2 Percolation models.

Connectedness of one phase in a composite material is often the principal feature of the mixture geometry which determines
effective behavior. For example, if highly conducting inclusions are sparsely distributed, forming a disconnected phase within
a poorly conducting encompassing host, then the effective conductivity will be poor as well. However, if there are enough
conducting inclusions so that they form connected pathways through the medium, then the effective conductivity will be much
closer to that of the inclusions. Percolation theory Broadbent and Hammersley (1957); Stauffer and Aharony (1992); Grimmett
(1989); Bunde and Havlin (1991) focuses on connectedness in disordered and inhomogeneous media, and has provided the
theoretical framework for describing the behavior of fluid flow through sea ice Golden et al. (1998a, 2007); Golden (2009).

Consider the $d-$dimensional integer lattice $\mathbb{Z}^d$, and the square or cubic network of bonds joining nearest neighbor lattice
sites. In the percolation model Broadbent and Hammersley (1957); Stauffer and Aharony (1992); Grimmett (1989); Bunde
and Havlin (1991), we assign to each bond a conductivity $\sigma_0 > 0$ with probability $p$, meaning it is open (black), and with
probability $1 - p$ we assign $\sigma_0 = 0$, meaning it is closed. Two examples of lattice configurations are shown in Fig. 2. with
$p = 1/3$ in (a) and $p = 2/3$ in (b). Groups of connected open bonds are called *open clusters*. In this model there is a critical
probability $p_c$, $0 < p_c < 1$, the *percolation threshold*, at which the average cluster size diverges and an infinite cluster appears.
For the $d = 2$ bond lattice $p_c = 1/2$. For $p < p_c$ the infinite cluster density $P_\infty(p) = 0$, while for $p > p_c$, $P_\infty(p) > 0$ and near
the threshold, $P_\infty(p) \sim (p - p_c)^\beta$ as $p \to p_c^+$, where $\beta$ is a universal critical exponent. It depends only on dimension and not
on the details of the lattice. Let $x, y \in \mathbb{Z}^d$ and $\tau(x, y)$ be the probability that $x$ and $y$ belong to the same open cluster. Then for
$p < p_c$, $\tau(x, y) \sim e^{-|x-y|/\xi(p)}$, and the correlation length $\xi(p) \sim (p_c - p)^{-\nu}$ diverges with a universal critical exponent $\nu$ as
$p \to p_c^-$. as shown in Fig. 2 (c).

The effective conductivity $\sigma^*(p)$ of the lattice, now viewed as a random resistor (or conductor) network, defined via Kir-
choff's laws, vanishes for $p < p_c$ like $P_\infty(p)$ since there are no infinite pathways. as shown in Fig. 2 (e). For $p > p_c$, $\sigma^*(p) > 0$,
and near $p_c$, $\sigma^*(p) \sim \sigma_0(p - p_c)^t$, $p \to p_c^+$, where $t$ is the conductivity critical exponent, with $1 \le t \le 2$ in $d = 2, 3$ Golden
(1990, 1992, 1997a), and numerical values $t \approx 1.3$ in $d = 2$ and $t \approx 2.0$ in $d = 3$ Stauffer and Aharony (1992). Consider a
random pipe network with fluid permeability $k^*(p)$ exhibiting similar behavior $k^*(p) \sim k_0(p - p_c)^e$, where $e$ is the perme-
ability critical exponent, with $e = t$ Chayes and Chayes (1986); Sahimi (1995); Golden (1997a). Both $t$ and $e$ are believed to
be universal – they depend only on dimension and not the lattice. Continuum models like the Swiss cheese model, can ex-
hibit nonuniversal behavior with exponents different from the lattice case and $e \ne t$ Halperin et al. (1985); Feng et al. (1987);
Stauffer and Aharony (1992); Sahimi (1994); Kerstein (1983).





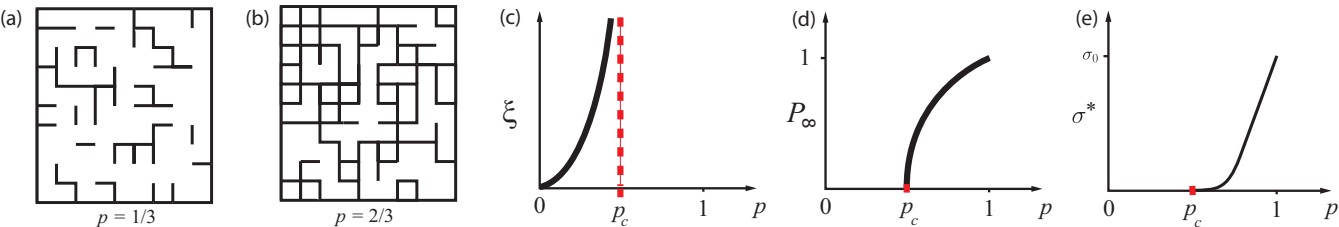

**Figure 2.** The two dimensional square lattice percolation model below its percolation threshold of $p_c = 1/2$ in (a) and above it in (b). (c) Divergence of the correlation length as $p$ approaches $p_c$. The infinite cluster density of the percolation model is shown in (d), and the effective conductivity is shown in (e).

## 3 Analytic continuation for two phase composites.

We now describe the *analytic continuation method* (ACM) for studying the effective properties of composites Bergman (1980); Milton (1980); Golden and Papanicolaou (1983); Golden (1997b). This method has been used to obtain rigorous bounds on bulk transport coefficients of composite materials from partial knowledge of the microstructure, such as the volume fractions of the phases. Examples of transport coefficients to which this approach applies include the complex permittivity, electrical and thermal conductivity, diffusivity, magnetic permeability, and elasticity. In Golden (1995); Golden et al. (1998c, b); Golden

(1997b, 2015, 2009); Golden et al. (2020) rigorous bounds on the complex permittivity of sea ice were found.

To set ideas we focus on complex permittivity. Consider a two-phase random medium with local permittivity tensor $\epsilon(x, \omega)$, a spatially stationary random field in $x \in \mathbb{R}^d$ and $\omega \in \Omega$, where $\Omega$ is the set of realizations of the medium. We consider a two-phase locally isotropic medium, where the components $\epsilon_{jk}$, $j, k = 1, .., d$, of $\epsilon$ satisfy

$$\epsilon_{jk}(x, \omega) = \epsilon(x, \omega) \delta_{jk}, \tag{1}$$

where $d$ is dimension, $\delta_{jk}$ is the Kronecker delta and

$$\epsilon(x, \omega) = \epsilon_1 \chi_1(x, \omega) + \epsilon_2 \chi_2(x, \omega). \tag{2}$$

Later, we will consider a polycrystalline medium where $\epsilon$ is a non-trivial symmetric matrix. Here $\chi_i(x, \omega)$ is the characteristic function of medium $i = 1, 2$, equaling 1 for $\omega \in \Omega$ with medium $i$ at $x$, and 0 otherwise, with $\chi_1 + \chi_2 = 1$. The random electric and displacement fields $E(x, \omega)$ and $D(x, \omega)$ satisfy

$$\nabla \times E = 0, \quad \nabla \cdot D = 0, \quad D = \epsilon E. \tag{3}$$

A variational problem establishes that $E$ can be written as $E = E_f + E_0$ satisfying

$$E = E_f + E_0, \qquad \nabla \times E_f = 0, \quad \langle D \cdot E_f \rangle = 0, \quad \langle E \rangle = E_0, \tag{4}$$

This basically amounts to saying curl-free and divergence-free fields are orthogonal (Helmholtz's theorem), but is rigorously established via the Lax-Milgram theorem Golden and Papanicolaou (1983).





The effective permittivity tensor $\boldsymbol{\epsilon}^*$ is defined as $\langle D \rangle = \boldsymbol{\epsilon}^* \langle E \rangle$, where $\langle \cdot \rangle$ is ensemble averaging over $\Omega$ or, by an ergodic theorem, spatial average over all of $\mathbb{R}^d$ Golden and Papanicolaou (1983). We prescribe that $E_0$ has direction $e_k$, the $k$th direction unit vector, and focus on the diagonal coefficient $\epsilon^* = \epsilon_{kk}^*$, with $\epsilon^* = \langle \epsilon E \cdot e_k \rangle$. The key step of the method is to obtain the following Stieltjes integral representation for $\epsilon^*$ Bergman (1978); Milton (1980); Golden and Papanicolaou (1983); Milton (2002),

$$F(s) = 1 - \frac{\epsilon^*}{\epsilon_2} = \int\limits_0^1 \frac{d\mu(\lambda)}{s - \lambda}, \qquad s = \frac{1}{1 - \epsilon_1/\epsilon_2}, \tag{5}$$

where $\mu$ is a positive Stieltjes measure on $[0,1]$. In the variable $h = \epsilon_1/\epsilon_2$, $F(s)$ is a *Stieltjes function* Golden (1997c); Cherkaev (2001); Murphy and Golden (2012). This formula arises from a resolvent formula for the electric field (in medium 1) Murphy et al. (2015),

$$\chi_1 E = s(sI - G)^{-1}\chi_1 e_k, \quad G = \chi_1 \Gamma \chi_1, \tag{6}$$

yielding $F(s) = \langle [(sI - G)^{-1}\chi_1 e_k] \cdot e_k \rangle$, where $\Gamma = -\nabla(-\Delta)^{-1}\nabla\cdot$ is a projection onto the range of the gradient operator $\nabla$ and $e_k$ is the standard basis vector in the $k$th direction. Formula (5) is the spectral representation of the resolvent and $\mu$ is the spectral measure of the self-adjoint operator $G = \chi_1 \Gamma \chi_1$ on $L^2(\Omega, P)$.

A critical feature of equation (5) is that the component parameters in $s$ are separated from the geometrical information in $\mu$. Information about the geometry enters through the moments

$$\mu_n = \int\limits_0^1 \lambda^n d\mu(\lambda) = \langle G^n \chi_1 e_k \cdot \chi_1 e_k \rangle. \tag{7}$$

Then $\mu_0 = \phi$, where $\phi$ is the volume or area fraction of phase 1, such as the brine volume fraction, the open water area fraction or melt pond coverage and $\mu_1 = \phi(1 - \phi)/d$ if the material is statistically isotropic. In general, $\mu_n$ depends on the $(n+1)$–point correlation function of the medium. This integral representation yields rigorous *forward bounds* for the effective parameters of composites, given partial information on the microgeometry via the $\mu_n$ Bergman (1980); Milton (1980); Golden and Papanicolaou (1983); Bergman (1982). One can also use the integral representations to obtain *inverse bounds*, allowing one to use data about the electromagnetic response of a sample, for example, to bound its structural parameters, such as the volume fraction of each of the components McPhedran et al. (1982); McPhedran and Milton (1990); Cherkaev and Golden (1998); Cherkaev (2001); Zhang and Cherkaev (2009); Bonifasi-Lista and Cherkaev (2009); Cherkaev and Bonifasi-Lista (2011); Day and Thorpe (1999); Golden et al. (2011), see Section 5 for more details.

### 230    3.1   Spectral measure computations for two phase composites

Computing the spectral measure $\mu$ for a given 2D composite microstructure geometry first involves discretizing a two phase image of the composite into a square lattice filled with 1's and 0's corresponding to the two phases. On this square lattice the action of the differential operators $\nabla$ and $\nabla\cdot$ are defined in terms of forward and backward difference operators Golden (1992).





Then the key operator $\chi_1 \Gamma \chi_1$, which depends on the geometry of the network via $\chi_1$, becomes a real-symmetric matrix $M$

Murphy et al. (2015). Here $\Gamma$ is a (non-random) projection matrix which depends only on the lattice topology and boundary conditions, and $\chi_1$ is a diagonal (random) projection matrix which determines the geometry and component connectivity of the composite medium Murphy et al. (2015).

The following theorem provides a rigorous mathematical formulation of integral representations for the effective parameters for finite lattice approximations of two component composite media. The electric field decomposition in this theorem is

established in Theorem 4 of Appendix A below and the integral representation in equation (8) is established in Theorem 2.1 of Murphy et al. (2015).

**Theorem 1.** *For each $\omega \in \Omega$, let $M(\omega) = W(\omega)\Lambda(\omega)W(\omega)$ be the eigenvalue decomposition of the real-symmetric matrix $M(\omega) = \chi_1(\omega)\Gamma\chi_1(\omega)$. Here, the columns of the matrix $W(\omega)$ consist of the orthonormal eigenvectors $\mathrm{w}_i(\omega)$, $i = 1,\ldots,N$, of $M(\omega)$ and the diagonal matrix $\Lambda(\omega) = \mathrm{diag}(\lambda_1(\omega),\ldots,\lambda_N(\omega))$ involves its eigenvalues $\lambda_i(\omega)$. Denote $Q_i = \mathrm{w}_i \mathrm{w}_i^T$ the*

*projection matrix onto the eigen-space spanned by $\mathrm{w}_i$ and denote $\delta_{\lambda_i}(\mathrm{d}\lambda)$ the Dirac $\delta$-measure centered at $\lambda_i$. The electric field $E(\omega)$ satisfies $E(\omega) = E_0 + E_f(\omega)$, with $E_0 = \langle E(\omega) \rangle$ and $\Gamma E(\omega) = E_f(\omega)$, and the effective complex permittivity tensor $\epsilon^*$ has components $\epsilon^*_{jk}$, $j,k = 1,\ldots,d$, which satisfy*

$$\epsilon^*_{jk} = \epsilon_2(\delta_{jk} - F_{jk}(s)), \qquad F_{jk}(s) = \int_0^1 \frac{\mathrm{d}\mu_{jk}(\lambda)}{s - \lambda}, \qquad \mathrm{d}\mu_{jk}(\lambda) = \sum_{i=1}^N \langle \delta_{\lambda_i}(\mathrm{d}\lambda)\, \chi_1 Q_i \hat{e}_j \cdot \hat{e}_k \rangle. \qquad (8)$$

From Theorem 1, the integral and $\chi_1 E$ in equations (5) and (6) have explicit representations in terms of the eigenvalues $\lambda_i$

and eigenvectors $u_i$ of $M$ Murphy et al. (2015),

$$\chi_1 E = s \sum_i \frac{\sqrt{m_i}}{s - \lambda_i}\, u_i, \qquad F(s) = \sum_i \left\langle \frac{m_i}{s - \lambda_i} \right\rangle, \qquad m_i = |\chi_1 u_i \cdot \hat{e}_k|^2, \qquad (9)$$

where $\hat{e}_k$ plays the role of a standard basis vector on the lattice. To compute $\mu$ a non-standard generalization of the spectral theorem for matrices is required, due to the projective nature of the matrices $\chi_1$ and $\Gamma$ Murphy et al. (2015). We developed

a *projection method* that shows the spectral measure $\mu$ in (8) depends only on the eigenvalues and eigenvectors of random sub-matrices of $\Gamma$ of size $N_1 \approx \phi N$ corresponding to diagonal components $[\chi_1]_{ii} = 1$, as the spectral weights $m_i$ (Christoffel numbers) associated with eigenvectors satisfying $\chi_1 u_i = 0$ are themselves zero, $m_i = 0$ Murphy et al. (2015). Fortunately, since these submatrices are much smaller for low volume fractions, this method greatly improves the efficiency and accuracy of numerical computations of $\mu$.

The measure $\mu$ exhibits fascinating transitional behavior as a function of system connectivity. For example, in the case of a RRN with a low volume fraction $p$ of open bonds, as shown in Fig. 2a, there are spectrum-free regions at the spectral endpoints $\lambda = 0, 1$ Murphy and Golden (2012). However, as $p$ approaches the percolation threshold $p_c$ Stauffer and Aharony (1992); Torquato (2002) and the system becomes increasingly connected, these spectral gaps shrink and then vanish Murphy and Golden (2012); Jonckheere and Luck (1998), leading to the formation of $\delta$-components of $\mu$ at the spectral endpoints,

*precisely* Murphy and Golden (2012) when $p = p_c$ (and $p = 1 - p_c$ in $d = 3$). This leads to critical behavior of $\sigma^*$ for insulating/conducting and conducting/superconducting systems Murphy and Golden (2012). This gap behavior of $\mu$ has led Golden

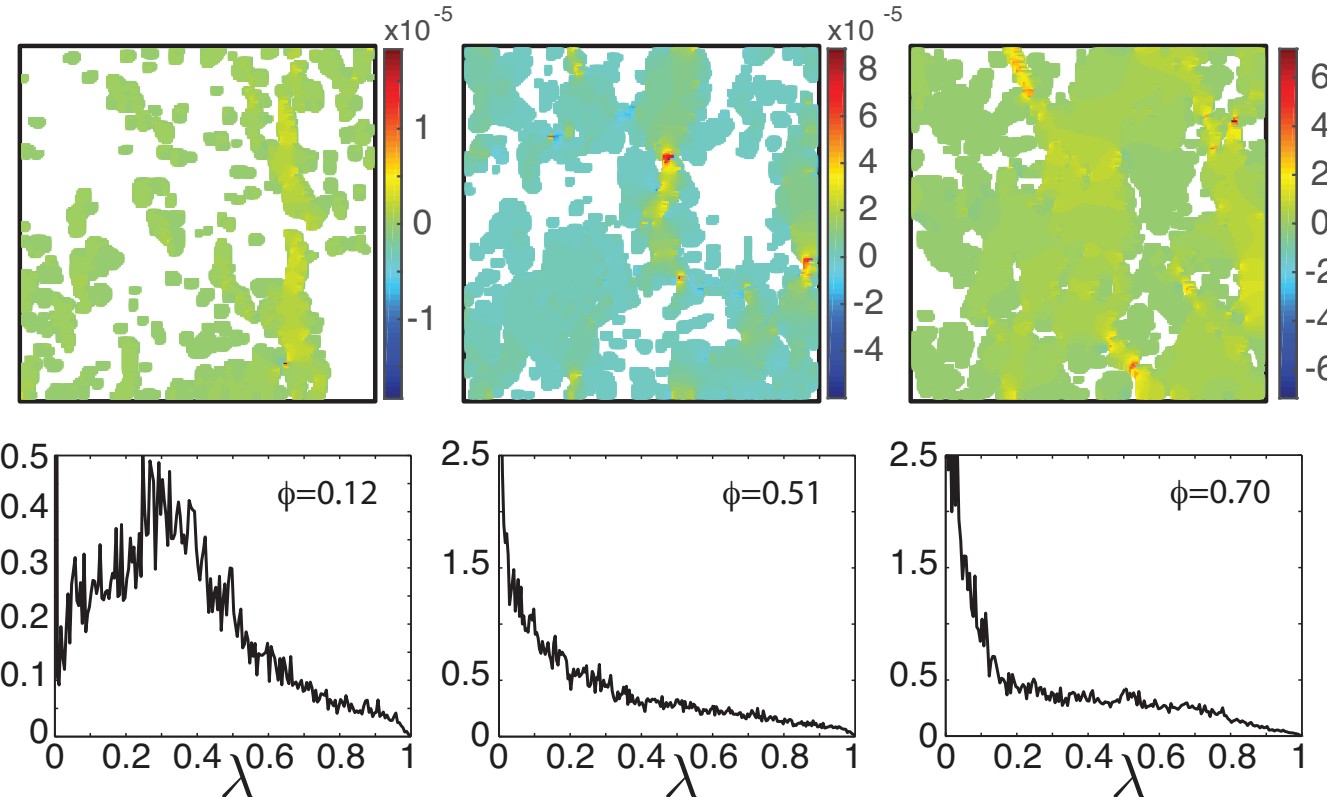

**Figure 3.** *Electric field and spectral function for sea ice brine microstructure.* Electric fields for X-ray CT images of 2D cross sections of 3D brine structures in sea ice (top) and corresponding spectral measures (bottom). As the brine fraction increases the fluid phase becomes increasingly connected and a delta function singularity in the spectral functions $\mu(\lambda)$ develops at $\lambda = 0$. This provides an electrical signature of brine connectivity, with a substantial increase in the strength of the electric field as the system attains global connectivity. Here, $E_0$ is taken to be vertically oriented.

(1997c); Murphy and Golden (2012) to a detailed description of these critical transitions in $\sigma^*$, which is analogous to the Lee–Yang–Ruelle–Baker description Baker (1990); Golden (1997c) of the Ising model phase transition in the magnetization $M$. Moreover, using this gap behavior, all of the classical critical exponent scaling relations were recovered Murphy and Golden (2012); Golden (1997c) without heuristic scaling forms Efros and Shklovskii (1976) but instead by using the *rigorous* integral representation for $\sigma^*$ involving $\mu$.

This spectral behavior emerges in all the systems mentioned above, such as the brine microstructure of sea ice Golden et al. (1998a, 2007); Golden (2009) as shown in Fig. 3, melt ponds on the surface of Arctic sea ice Hohenegger et al. (2012) as shown in Fig. 4, and the sea ice pack itself Murphy et al. (2017a). This also gives rise to critical behavior of the electric field as shown in Fig. 3 for 2D cross sections of 3D brine microstructure, with $E_0$ taken to be in the vertical direction. Disconnected and weakly connected examples of brine microstructure have small values of the electric field, while strongly connected

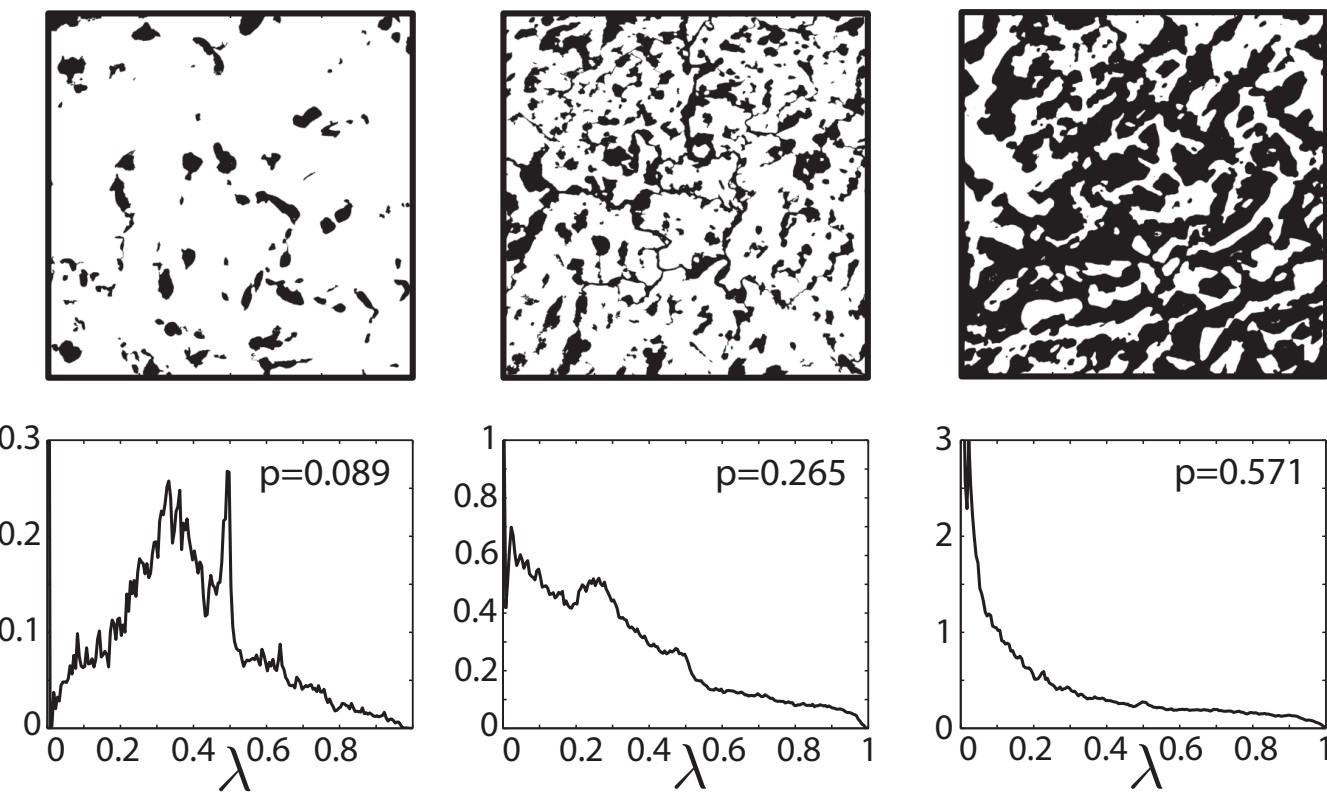

**Figure 4.** *Sea ice melt ponds.* Melt ponds on the surface of the sea ice (top) (images courtesy of Don Perovich) and corresponding spectral functions (bottom). As the melt pond area fraction increases the ice/water composites become increasingly connected and a delta function singularity in the spectral functions $\mu(\lambda)$ develops at $\lambda = 0$.

brine microstructures are characterized by a substantial increase in the strength of the electric field. A similar behavior of the temperature gradient $\nabla T$ associated with the Stieltjes integral for the horizontal thermal conductivity of melt ponds atop Arctic sea ice is shown in Fig. 4.

## 3.2 Generalization to rank deficient setting

In the periodic setting, for example, the matrix Laplacian is singular so the matrix representation of $(-\Delta)^{-1}$ in $\Gamma$ is not defined. We now extend the mathematical framework developed in Murphy et al. (2015) to this setting. To make the connection to the abstract Hilbert space Golden and Papanicolaou (1983) and full rank matrix Murphy et al. (2015) settings, we first give relevant details for these cases. Equation (6) for the abstract Hilbert space setting follows by applying the operator $-\nabla(-\Delta)^{-1}$ to the formula $\nabla \cdot D = 0$, yielding $\Gamma D = 0$. Equation (6) then follows by using $\Gamma E_f = E_f$ and $\Gamma E_0 = 0$ Murphy et al. (2015), since $E_f$ is in the range of $\Gamma$ and $E_0$ is constant Murphy et al. (2020, 2017b, 2015). The matrix form of $\nabla \cdot D = 0$ is $-\nabla^T D = 0$, where $\nabla$ now represents the finite difference matrix representation of the gradient operator and $-\nabla^T$ is the





finite difference representation of the divergence operator, with *negative* matrix Laplacian given by $\nabla^T \nabla$ Murphy et al. (2015).
As before, in Murphy et al. (2015) we applied the matrix $\nabla(\nabla^T \nabla)^{-1}$ to the formula $-\nabla^T D = 0$, yielding $\Gamma D = 0$, where

$\Gamma = \nabla(\nabla^T \nabla)^{-1}\nabla^T$, and equation (6) follows the same way as before.

Now consider the singular value decomposition of the matrix gradient Murphy et al. (2020) of size $m \times n$, say, $\nabla = U\Sigma V^T$.
Here $U$ is a $m \times n$ matrix satisfying $U^T U = I_n$, $\Sigma$ is a $n \times n$ diagonal matrix with diagonal entries consisting of the *singular values* of $\nabla$, and $V$ is a $n \times n$ orthogonal matrix satisfying $V^T V = VV^T = I_n$, where $I_n$ is the identity matrix of size $n$. When the matrix gradient is full rank it has $n$ strictly positive singular values, so $\Sigma$ is an invertible matrix and the matrix representation

of $\Gamma$ is given by $\Gamma = UU^T$. On the other hand, when the matrix gradient is singular we have $\Sigma = \text{diag}(\Sigma_1, 0, \ldots, 0)$, where the diagonal matrix $\Sigma_1$ contains the $n_1$ strictly positive singular values of $\Sigma$ and the rest of the singular values have value 0. Denoting $U_1$ and $V_1$ to be the columns of $U$ and $V$ corresponding to the diagonal entries of $\Sigma_1$, we have $\nabla = U_1 \Sigma_1 V_1^T$, where $\Sigma_1$ is invertable and $U_1^T U_1 = V_1^T V_1 = I_{n_1}$. This enables us to write $-\nabla^T D = 0$ as $-V_1 \Sigma_1 U_1^T D = 0$, hence $U_1^T D = 0$ and $U_1 U_1^T D = 0$. Noting that the columns of $U_1$ span the range of the matrix gradient $\nabla$, the matrix $U_1 U_1^T$ is a projection onto the

range of $\nabla$ Murphy et al. (2020). Defining $\Gamma = U_1 U_1^T$, equation (6) follows the same way as before. This clearly generalizes the full rank setting. More details are given in the appendix in Section A.

## 4 Analytic continuation for polycrystalline media

Sea ice is a composite material with polycrystalline microstructure on the millimeter to centimeter scale. When sea water freezes under turbulent forcing, granular sea ice forms, having small crystals with isotropic orientation angles. Columnar sea

ice forms in quiescent conditions, with large crystals more strongly oriented in the vertical direction. Examples of granular and columnar sea ice polycrystal microgeometry are displayed in Fig. 5 (a) and (d).

Our analysis of the transport properties of random, uniaxial polycrystalline media Barabash and Stroud (1999) in Gully et al. (2015), and a somewhat new formulation presented below, shows the underlying mathematical framework is a direct analogue of that for two-phase random media discussed in Sec. 3. For simplicity, we discuss electrical permittivity $\epsilon$, keeping in mind the

broader applicability to thermal conductivity $\kappa$, electric conductivity $\sigma$, etc. Polycrystalline materials, are composed of many crystallites (single crystals of varying size, shape, and orientation) that can have different local conductivities along different crystal axes. In contrast to equation (1), the local permittivity matrix of such media is given by Milton (2002); Barabash and Stroud (1999)

$$\epsilon(x, \omega) = R^T \Phi R, \quad \Phi = \text{diag}(\epsilon_1, \ldots, \epsilon_d), \tag{10}$$

where $R(x, \omega)$ is a random rotation matrix satisfying $R^T = R^{-1}$. For example, for $d = 2$ we have

$$\epsilon = R^T \begin{bmatrix} \epsilon_1 & 0 \\ 0 & \epsilon_2 \end{bmatrix} R, \qquad R = \begin{bmatrix} \cos\theta & -\sin\theta \\ \sin\theta & \cos\theta \end{bmatrix}, \tag{11}$$

where $\theta = \theta(x, \omega)$ is the orientation angle, measured from the direction $e_1$, of the polycrystallite which has an interior containing $x \in \mathbb{R}^d$ for $\omega \in \Omega$. In higher dimensions, $d \geq 3$, the rotation matrix $R$ is a composition of "basic" rotation matrices $R_i$, e.g.




$R = \prod_{j=1}^{d} R_j$, where the matrix $R_j(x,\omega)$ rotates vectors in $\mathbb{R}^d$ by an angle $\theta_j = \theta_j(x,\omega)$ about the $e_j$ axis. For example, in three dimensions

$$R_1 = \begin{bmatrix} 1 & 0 & 0 \\ 0 & \cos\theta_1 & -\sin\theta_1 \\ 0 & \sin\theta_1 & \cos\theta_1 \end{bmatrix}, \quad R_2 = \begin{bmatrix} \cos\theta_2 & 0 & \sin\theta_2 \\ 0 & 1 & 0 \\ -\sin\theta_2 & 0 & \cos\theta_2 \end{bmatrix}, \quad R_3 = \begin{bmatrix} \cos\theta_3 & -\sin\theta_3 & 0 \\ \sin\theta_3 & \cos\theta_3 & 0 \\ 0 & 0 & 1 \end{bmatrix}. \tag{12}$$

In the case of *uniaxial* polycrystalline media, the local permittivity along one of the crystal axes has the value $\epsilon_1$, while the permittivity along all the other crystal axes has the value $\epsilon_2$, so $\Phi = \mathrm{diag}(\epsilon_1, \epsilon_2)$ for 2D (which is the general setting for 2D) and $\Phi = \mathrm{diag}(\epsilon_1, \epsilon_2, \epsilon_2)$ for 3D. Equation (10) can be written in a more suggestive form in terms of the matrix $C = \mathrm{diag}(1, 0, \ldots, 0)$

$$\epsilon(x,\omega) = \epsilon_1 X_1(x,\omega) + \epsilon_2 X_2(x,\omega), \tag{13}$$

which is an analogue of equation (2). Here $X_1 = R^T C R$ and $X_2 = R^T(I - C)R$, where $I$ is the identity matrix on $\mathbb{R}^d$. Since $R^T = R^{-1}$ and $C$ is a diagonal projection matrix satisfying $C^2 = C$, it is clear that the $X_i$, $i = 1, 2$, are mutually orthogonal projection matrices satisfying

$$X_j^T = X_j, \quad X_j X_k = X_j \delta_{jk}, \quad X_1 + X_2 = I, \tag{14}$$

which are also properties of the characteristic functions $\chi_j$ in Sec. 3.

Equations (3) and (4) are also satisfied in this polycrystalline setting Golden and Papanicolaou (1983). Similar to the derivation of equation (6) in Sec. 3, a resolvent representation for $X_1 E$ follows by applying the operator $-\nabla(-\Delta)^{-1}$ to the formula $\nabla \cdot D = 0$, yielding $\Gamma D = 0$. Then, using $\Gamma E_f = E_f$ and $\Gamma E_0 = 0$ Murphy et al. (2015) yields the following analogue of equation (6)

$$X_1 E = s(sI - G)^{-1} X_1 e_k, \quad G = X_1 \Gamma X_1, \tag{15}$$

yielding the integral representation in equation (5) for $F(s) = \langle [(sI - G)^{-1} X_1 e_k] \cdot e_k \rangle$. As in the two component setting, a critical feature of equation (5) is that the component parameters in $s$ are separated from the geometrical information in $\mu$. Information about the geometry enters through the moments in equation (7) with $G$ given in (15) and $\chi_1$ replaced by $X_1$. The mass $\mu_0$ of the measure $\mu_{jk}$ is given by

$$\mu_{jk}^0 = \langle X_1 e_j \cdot e_k \rangle, \quad \mu_{kk}^0 = \langle |X_1 e_k|^2 \rangle, \tag{16}$$

where the second equality follows from the fact that $X_1$ is a real-symmetric projection matrix. The statistical average $\langle |X_1 e_k|^2 \rangle$ in (16) can be thought of as the "mean orientation," or as the percentage of crystallites oriented in the $k^{\text{th}}$ direction. For example, in the case of two-dimensional polycrystalline media, $d = 2$, equation (11) implies that

$$\mu_{11}^0 = \langle \cos^2 \theta \rangle, \quad \mu_{22}^0 = \langle \sin^2 \theta \rangle, \quad \mu_{12}^0 = \langle \sin\theta \cos\theta \rangle. \tag{17}$$

Generalizing equation (12), with $R = \prod_{j=1}^{d} R_j$, to dimensions $d \geq 3$ shows that $\mu_{jk}^0$ is a linear combination of averages of the form $\langle \prod_i \cos^{n_i} \theta_i \sin^{m_i} \theta_i \rangle$, where $n_i, m_i = 0, 1, 2, \ldots$.

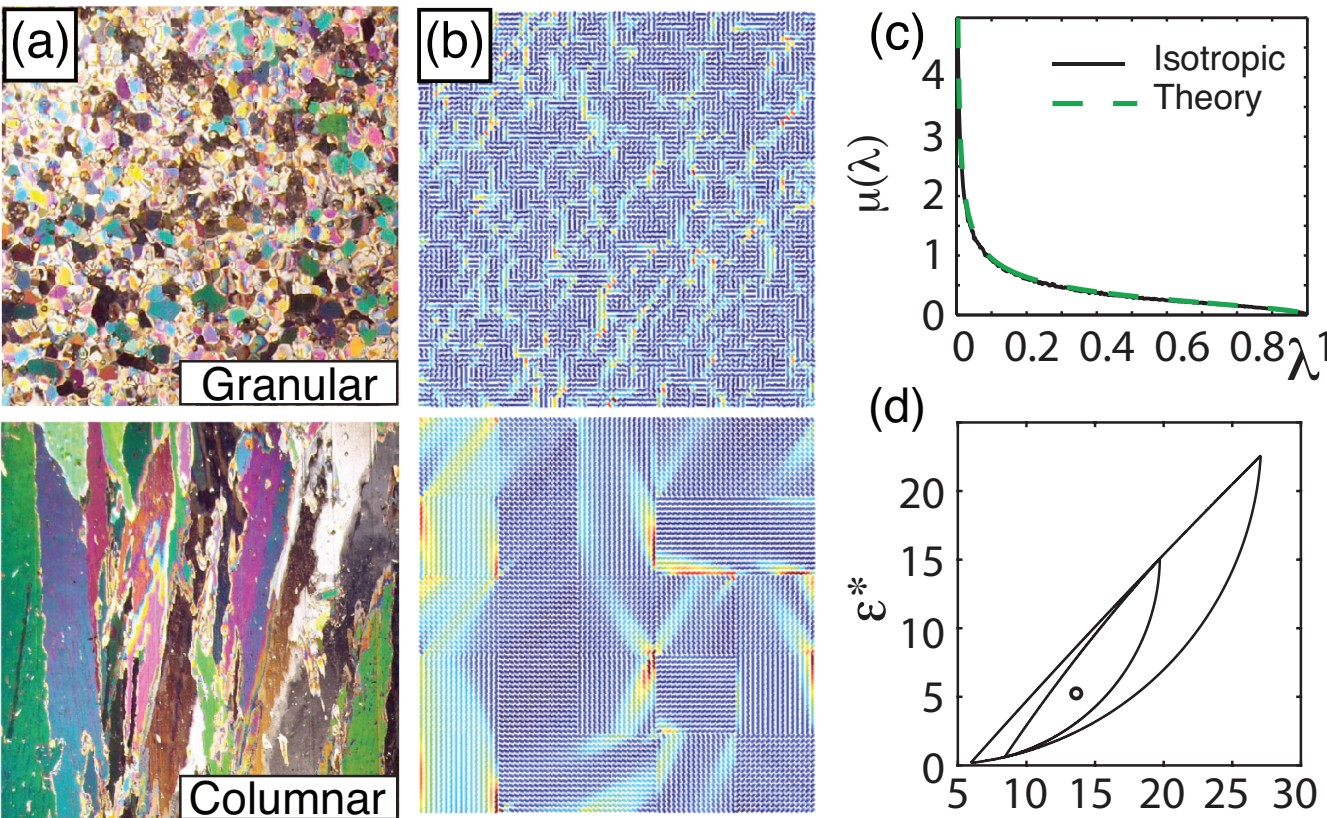

**Figure 5.** *Spectral analysis of polycrystalline media.* (a) Cross sections of polycrystalline microstructure for granular and columnar sea ice. (b) Discrete checkerboard polycrystal microstructure with isotropic crystallite orientations within the horizontal plane, with small (top) and large (bottom) crystallite size. Cool and warm colors correspond to low and high displacement fields. (c) The spectral function, a histogram representation of the spectral measure $\mu$ shown along with it's theoretical prediction for such isotropic media Milton (2002). (d) An example value of the complex effective permittivity of isotropic polycrystalline media captured by first and second order bounds Gully et al. (2015).

The integral representation (5) for this polycrystalline setting yields rigorous *forward bounds* for the effective parameters of composites, given partial information on the microgeometry via the $\mu_n$ Gully et al. (2015); Milton (2002), as shown in Fig. 5d below. One can also use the integral representations to obtain *inverse bounds*, allowing one to use data about the electromagnetic response of a sample, for example, to bound its structural parameters, such as the average crystallite orientation Gully et al. (2015); Milton (2002), see Section 5 for more details.

### 4.1 Spectral measure computations for uniaxial polycrystalline materials

Computing the spectral measure $\mu$ for a given polycrystalline microgeometry first involves discretizing the composite into a square lattice with vertex values in the range $[0, 2\pi]$ corresponding to the crystallite orientation angles at each vertex location.





On this square lattice the action of the differential operators $\nabla$ and $\nabla\cdot$ are defined in terms of forward and backward difference operators Golden (1992). Then the key operator $X_1 \Gamma X_1$, which depends on the geometry of the network via $X_1$, becomes a real-symmetric matrix $M$. Here $\Gamma$ is as in Sec. 3.1 and $X_1$ is a banded (random) projection matrix which determines the geometry of the polycrystalline medium. In this setting, the integral and $X_1 E$ in equations (5) and (6) have explicit representations in terms of the eigenvalues $\lambda_i$ and eigenvectors $u_i$ of $M$ Murphy et al. (2015) given by equation (9), and similarly the spectral measure is given by equation (8), with $\chi_1$ replaced by $X_1$.

The following theorem provides a rigorous mathematical formulation of integral representations for the effective parameters for finite lattice approximations of random uniaxial polycrystaline media.

**Theorem 2.** *For each $\omega \in \Omega$, let $M(\omega) = W(\omega)\Lambda(\omega)W(\omega)$ be the eigenvalue decomposition of the real-symmetric matrix $M(\omega) = X_1(\omega)\Gamma X_1(\omega)$. Here, the columns of the matrix $W(\omega)$ consist of the orthonormal eigenvectors $\mathrm{w}_i(\omega)$, $i = 1, \ldots, N$, of $M(\omega)$ and the diagonal matrix $\Lambda(\omega) = \mathrm{diag}(\lambda_1(\omega), \ldots, \lambda_N(\omega))$ involves its eigenvalues $\lambda_i(\omega)$. Denote $Q_i = \mathrm{w}_i \mathrm{w}_i^T$ the projection matrix onto the eigen-space spanned by $\mathrm{w}_i$. The electric field $E(\omega)$ satisfies $E(\omega) = E_0 + E_f(\omega)$, with $E_0 = \langle E(\omega) \rangle$ and $\Gamma E(\omega) = E_f(\omega)$, and the effective complex permittivity tensor $\boldsymbol{\epsilon}^*$ has components $\epsilon_{jk}^*$, $j, k = 1, \ldots, d$, which satisfy*

$$\epsilon_{jk}^* = \epsilon_2(\delta_{jk} - F_{jk}(s)), \qquad F_{jk}(s) = \int_0^1 \frac{\mathrm{d}\mu_{jk}(\lambda)}{s - \lambda}, \qquad \mathrm{d}\mu_{jk}(\lambda) = \sum_{i=1}^N \langle \delta_{\lambda_i}(\mathrm{d}\lambda) X_1 Q_i \hat{e}_j \cdot \hat{e}_k \rangle. \qquad (18)$$

We defer the proof of Theorem 2 to Section B, which holds for both of the settings where the matrix gradient is full rank or rank deficient. To numerically compute $\mu$ a non-standard generalization of the spectral theorem for matrices is required, due to the projective nature of the matrices $X_1$ and $\Gamma$ Murphy et al. (2015). In particular, in Section B we develop a *projection method* that shows the spectral measure $\mu$ in (18) depends only on the eigenvalues and eigenvectors of the upper left $N_1 \times N_1$ block of the matrix $R\Gamma R^T$, where $N_1 = N/d$. These submatrices are smaller by a factor of $d$, which improves the efficiency and numerical computations of $\mu$ by a factor of $d^3$.

In Fig. 5 computations of the displacement field $D$ are displayed for 2D polycrystaline media for small and large crystal sizes, along side cross sections of polycrystalline microstructure for granular and columnar sea ice. When the effective permittivity tensor $\boldsymbol{\epsilon}^*$ is diagonal, such as the setting of isotropically oriented crystallites, the spectral measure for an infinite system is known in closed form Milton (2002) to be $d\mu(\lambda) = (\sqrt{(1 - \lambda)/\lambda})(d\lambda/\pi)$, as shown in Fig. 5 (c). This measure has a singularity at $\lambda = 0$, which indicates that the material is electrically conductive, on macroscopic length scales Murphy et al. (2015); Murphy and Golden (2012). When the polycrystalline material has isotropic oriented crystallite angles, both the mass and first moment of the measure $\mu$ are known, which enables two nested bounds for $\epsilon$ to be computed Gully et al. (2015), as shown in Fig. 5 (d).

## 5 Inverse homogenization: Inverse problem of recovery information about the structure of composites

Developed originally for the effective complex permittivity $\epsilon^*$, the integral representation (5) yields rigorous *forward bounds* for the effective permittivity $\epsilon^*$ of two-component composites formed of materials with permittivity $\epsilon_1$ and $\epsilon_2$, given partial


information on the microgeometry via the moments $\mu_n$ Bergman (1980); Milton (1980); Bergman (1982); Golden and Papanicolaou (1983). One can also use the integral representation to recover information about the structure of composite material, this is the problem of **inverse homogenization**. For the inverse homogenization, it is important that the representation (5) separates information about the properties of the phases contained in the parameter $s$ from information about the microgeometry contained in the measure $\mu$ and its moments $\mu_n = \langle G^n \chi_1 e_k \cdot \chi_1 e_k \rangle$ (7) via higher-order correlation functions of the geometry function $\chi_1$.

Spectral measure $\mu$ and its moments $\mu_n$ contain, in principle, all the geometrical information about the composite. For example, the mass $\mu_0$ is the volume fraction $\phi$ of the first component in the composite,

$$\mu_0 = \int\limits_0^1 d\mu(z) = \langle \chi_1 \rangle = \phi, \tag{19}$$

and the fraction of the second phase is $1 - \phi$. Connectivity information is also embedded in the spectral measure.

The basis for inverse homogenization is provided by the uniqueness theorem Cherkaev (2001) which formulates the conditions under which the measure $\mu$ in the representation (5) can be uniquely reconstructed from measured data. For instance, electromagnetic data measured for a range of frequency of the applied electromagnetic field, are sufficient to uniquely recover the measure $\mu$ in (5). Such data are also sufficient for unique reconstruction of the moments $\mu_n$ Cherkaev and Ou (2008), provided the permittivity of one of the phases is frequency dependent. Two major approaches to the inverse homogenization are the *reconstruction of the measure $\mu$* Cherkaev (2001); Cherkaev and Ou (2008); Day and Thorpe (1996); Zhang and Cherkaev (2009); Bonifasi-Lista and Cherkaev (2009); Bonifasi-Lista et al. (2009); Cherkaev and Bonifasi-Lista (2011); Day and Thorpe (1999); Day et al. (2000); Golden et al. (2011); Cherkaev (2020) (and then calculating its moments) and *inverse bounds* for the structural parameters, such as, for example, the volume fraction of each of the components McPhedran et al. (1982); McPhedran and Milton (1990); Cherkaev and Tripp (1996); Cherkaev and Golden (1998); Cherkaev (2001); Cherkaev and Ou (2008), orientation of the crystals Gully et al. (2015) or connectedness Orum et al. (2012) of the structure.

When only a few data points are available, though the uniqueness theorem Cherkaev (2001) is not immediately applicable, one can outline a set of measures consistent with the measurements,

$$\mathcal{M} = \{\mu : F_\mu(s) = 1 - \epsilon^*/\epsilon_2\}, \tag{20}$$

and determine an interval confining the first moment of the measure $\mu$ providing, for instance, an interval of uncertainty for the volume fraction of one material. For several data points corresponding to the same structure of the composite, such as for example, measurements at a few different frequencies, the bounds for the volume fraction are given by an intersection of all admissible intervals Cherkaev and Tripp (1996); Cherkaev and Golden (1998); Tripp et al. (1998). When the requirements for the measurements needed to uniquely reconstruct the spectral measure $\mu$ established by the uniqueness theorem are satisfied, the set $\mathcal{M}$ is reduced to one point. But the map from the set of measures to the set of the microgeometries is not unique, and there is a variety of microstructures generating the same response under the applied field. Different microgeometries corresponding to the same sequence of moments $\mu_0, \mu_1, ...$ are the $S-$equivalent structures Cherkaev (2001) that are not distinguishable by homogenized measurements.





An equivalent representation for function $F(s)$ in (5) using a logarithmic potential of the measure $\mu$ on the complex plane
of variable $s$ is Cherkaev (2001):

$$F(s) = \frac{\partial}{\partial s} \int \ln|s - z|\, d\mu(z), \qquad \partial/\partial s = (\partial/\partial x - i\,\partial/\partial y), \qquad s = \frac{1}{1 - \epsilon_1/\epsilon_2}. \tag{21}$$

The solution to the inverse problem of recovering the measure $\mu$ is constructed solving the minimization problem:

$$\min_\mu \|A\mu - F\|^2, \qquad F(s) = 1 - \epsilon^*(s)/\epsilon_2 \tag{22}$$

where $A$ is the integral operator in (21) or in (5), the norm is the $L^2-$norm, $F = F(s)$, $s \in \mathbb{C}$, is the given function of the
measured data, and $\mathbb{C}$ is a curve on the complex plane corresponding to the frequencies of the applied field. The solution of the
minimization problem does not depend continuously on the data. Unboundedness of the operator $A^{-1}$ leads to arbitrarily large
variations in the solution, and the problem requires regularization to design a stable numerical algorithm Cherkaev (2001).
Regularized inversion schemes and stable reconstruction algorithms to recover $\mu$ and its moments from data on the effective
complex permittivity were developed in Cherkaev (2001, 2004); Cherkaev and Ou (2008); Bonifasi-Lista and Cherkaev (2009);
Cherkaev and Bonifasi-Lista (2011) based on $L^2, TV$, and non-negativity constraints, and constrained Pade approximation of
the measure $\mu$ Zhang and Cherkaev (2009). In application to imaging of bone structure, spectral measures $\mu$ computed with the
regularization algorithms based on $L^2$ constrained minimization, from electromagnetic Bonifasi-Lista and Cherkaev (2009);
Cherkaev and Bonifasi-Lista (2011); Golden et al. (2011) and viscoelastic Bonifasi-Lista and Cherkaev (2008); Bonifasi-Lista
et al. (2009); Cherkaev and Bonifasi-Lista (2011) data allow to distinguish the samples of healthy and osteoporotic bone via
the different microstructures and the connectivity of the trabecular architecture.

With hydrostatic and deviatoric projections $\Lambda_h$ and $\Lambda_s$ onto the orthogonal subspaces of the second order tensors comprised
of tensors proportional to the identity tensor and trace-free tensors, the Stieljtes integral representation was generalized in
Cherkaev and Bonifasi-Lista (2011) to the effective viscoelastic modulus and to two-dimensional viscoelastic polycrystalline
materials Cherkaev (2019) under the assumption that the constituents have the same elastic bulk and different (elastic and
viscoelastic) shear moduli. This representation was also used in inverse homogenization Bonifasi-Lista and Cherkaev (2008);
Cherkaev and Bonifasi-Lista (2011); Cherkaev (2020) for successful recovering the porosity of a composite from known
viscoelastic shear modulus.

Other approaches to the volume fraction bounds include Engström (2005); Milton (2012); Thaler and Milton (2014) based
on estimates for higher order moments and on variational bounds, as well as direct inversion of known formulas or mixing rules
Bergman and Stroud (1992); Levy and Cherkaev (2013) for effective properties of composites with specific structure, however,
an advantage of the methods discussed here, is their applicability without a priori assumption about the microgeometry.

**Spectral coupling of various properties of composites.** An important application of inverse homogenization is for indirect evaluating properties of materials through cross-coupling Milton (2002). Different properties of composites are coupled
through their microgeometry; this phenomenon has been known for a long time and used for estimating difficult to measure
directly properties, from available data. The conventional approaches are based on empirical and semi-empirical relations, such
as for instance, Kozeny-Carman or Katz-Tompson. These relations estimate permeability of a porous material characterizing





the microstructure by a "formation factor" $F$ which relates properties of one phase in the composite to the effective properties of the material.

In the *spectral coupling* method Cherkaev (2001) based on properties of the Stieltjes representation (5), the spectral measure $\mu$ is associated with the geometric structural function as this is the function that couples various properties of the same material. The method of spectral coupling Cherkaev (2001, 2004); Cherkaev and Zhang (2003); Cherkaev and Bonifasi-Lista (2011) for two component composites based on this coupling of different properties of the composite through the spectral measure allows us to recover various transport properties of sea ice from the spectral measures computed using other measured properties. In particular, this approach results in an indirect method of calculation of the thermal conductivity Cherkaev and Zhang (2003) and hydraulic conductivity of polycrystalline sea ice, difficult to measure over large scales, from the effective complex permittivity data (recovered from radar measurements). The spectral coupling was extended to evaluating viscoelastic properties of two component composite in Cherkaev and Bonifasi-Lista (2011) in application to characterizing bone properties and microarchitecture.

Inverse homogenization for recovering microstructural parameters from effective property measurements is applicable to problems in remote sensing, medical imaging, non-destructive testing of materials, and allows for example, to use Synthetic Aperture Radar (SAR) remote sensing for assessing the structure and transport properties of sea ice.

### 5.1 Bounds for the moments of the spectral measure

The second approach to the inverse homogenization problem is calculating *inverse bounds* for the structural parameters, such as, for example, the volume fraction of each of the components McPhedran et al. (1982); McPhedran and Milton (1990); Cherkaev and Tripp (1996); Cherkaev and Golden (1998); Cherkaev (2001), orientation of the crystals Gully et al. (2015) or connectedness Orum et al. (2012) of the structure. An analytical approach to estimating the volume fractions of materials in a composite Cherkaev and Tripp (1996); Cherkaev and Golden (1998); Tripp et al. (1998) gives explicit analytic formulas for the first order inverse bounds on the volume fractions of the constituents in a general composite and second order inverse bounds on the fractions of the phases in an isotropic composite Cherkaev and Golden (1998).

The inverse bounds are derived using analyticity of the effective complex permittivity of the composite. The first order bounds $p_l^{(1)}$ and $p_u^{(1)}$ for the volume fraction $\phi$ give the lower and upper bounds for the zero moment $\mu_0$ of the measure $\mu$ or its mass in (19) Cherkaev and Tripp (1996); Cherkaev and Golden (1998):

$$p_l^{(1)} \le \phi \le p_u^{(1)}, \qquad p_l^{(1)} = |f|^2 \frac{Im\,(\bar{s})}{Im\,(f)}, \qquad p_u^{(1)} = 1 - \frac{|g|^2\,Im\,(\bar{t})}{Im\,(g)}. \tag{23}$$

Here $t = 1 - s$, $f$ is the known value of $F(s)$, and $g$ is the known value of $G(t) = 1 - \epsilon^*/\epsilon_1$.

First and second order forward and inverse bounds are illustrated in Fig. 6(a) Cherkaev and Golden (1998) where first order bounds for the effective complex permittivity of all anisotropic composites that could be formed from two materials of permittivity $\epsilon_1$ and $\epsilon_2$ are presented in the left panel, while the second order isotropic bounds are shown in right panel. The small lens shaped domains each contain the anisotropic (left) and isotropic (right) mixtures corresponding to the volume fractions $\phi$ of the first component equal to $p_l^{(q)}$ and $p_u^{(q)}$, $q = 1, 2$. The points $p_l^{(q)}$ and $p_u^{(q)}$ give the lower and upper bounds


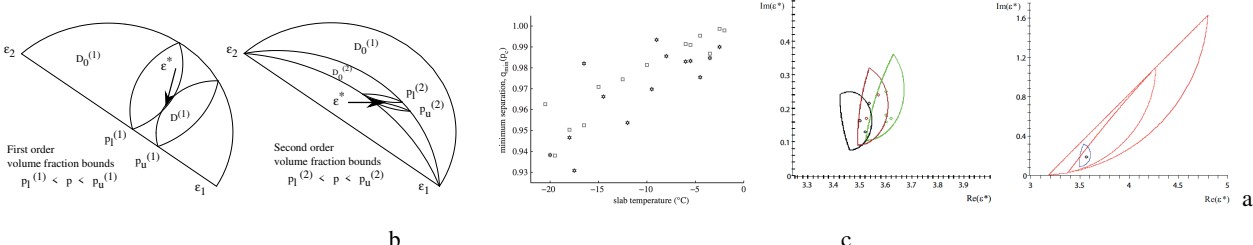

**Figure 6.** *Forward and Inverse bounds.* (a). Illustration of bounds on the volume fraction of one component in the mixture derived from first order anisotropic bounds (left panel), and from the second order isotropic bounds (right panel) for the effective permittivity Cherkaev and Golden (1998). The small lens shaped domains each contain $\epsilon^*$ of the anisotropic (left) and isotropic (right) composites corresponding to the volume fractions of the first component $p_l$ and $p_u$ which give the lower and upper bounds for the fraction of the first material. (b). Center figure shows lower bounds on separation parameter $q_{min}$ versus temperature Orum et al. (2012), calculated using data of the effective complex permittivity. The inverted data clearly indicate that as the ice warms, the separations of the brine inclusions decrease. Stars and squares indicate different sea ice slabs. (c). Polycrystalline bounds Gully et al. (2015) for the permittivity sea ice (left) together with the measured effective permittivity of sea ice in Arcone et al. (1986). Comparison of the polycrystalline bounds with the two-component bounds (right) shows a dramatic improvement over the classic two-component bounds as the new bounds include additional information about single crystal orientations. (Notice very different scales on the axes.)

for the volume fraction of the first material in the composite. Superscripts $q = 1$ and $q = 2$ indicate the first and second order bounds.

For a set of data points $\epsilon^*(k)$, $k = 1, ..., N$, corresponding to the same structure the bounds for the fraction $\phi$ of the first phase in the composite are given by an intersection of all admissible intervals $p_l^{(q)}(k) \leq \phi \leq p_u^{(q)}(k)$:

$$P_l^{(q)} = \max_k p_l^{(q)}(k) \leq \phi \leq \min_k p_u^{(q)}(k) = P_u^{(q)}, \qquad q = 1, 2. \tag{24}$$

Here $p_l^{(q)}(k)$ and $p_u^{(q)}(k)$ are, respectively, lower and upper bounds for the volume fraction $\phi$ calculated using the effective complex permittivity $\epsilon^*(k)$, and $q$ is the order of the bounds, $q = 1$ for a general mixture, $q = 2$ for an isotropic composite.

In Cherkaev and Golden (1998) this method was applied to estimating brine volume in sea ice from two data sets of 4.75 GHz measurements of the complex permittivity $\epsilon^*$ of sea ice Arcone et al. (1986) at $-6°$C and $-11°$C with fractions of brine $\phi = 0.036$ and $\phi = 0.0205$. Sea ice was considered as a composite of three components: pure ice, brine, and air; the effective

complex permittivity of the mixture of ice and air was calculated with the Maxwell Garnett formula. The first order bounds estimate the brine volume fraction as $0.0213 \leq \phi \leq 0.0664$ and $0.0119 \leq \phi \leq 0.0320$, for the data set 1 and 2, respectively. The second order inverse bounds derived with the assumption of 2D isotropy in the horizontal plane give the following estimates for the brine volume fraction: $0.0333 \leq \phi \leq 0.0422$ for the first data set with brine volume $\phi = 0.036$, and $0.0189 \leq \phi \leq 0.0213$ for the second data set with volume fraction of brine $\phi = 0.0205$.

First order bounds are further extended to polycrystalline materials and allow to estimate the mean crystal orientation Gully et al. (2015).

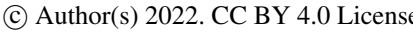



## 5.2 Matrix particle forward and inverse bounds

Another parameter important in characterizing the structure of composite material consisting of inclusions within a host matrix, is separation between the inclusions. Inclusion separation is an indicator of connectedness of phases – a key feature in critical
behavior and phase transitions; the separation parameter may be used to estimate closeness to the percolation phase transition.

Composites with non-touching inclusions of one material embedded in a host matrix of different material are called matrix particle composites. For a matrix particle composite with separated inclusions tighter bounds on the effective complex permittivity may be obtained. In Orum et al. (2012) sea-ice is considered as a matrix particle composite in which the brine phase contained in separated, circular discs of radii $r_b$ randomly located on a horizontal plane, is surrounded by a "corona" of
ice, with outer radius $r_i$. Such a material is called a $q$-material, where $q = r_b/r_i$. The minimal separation of brine inclusions is $2(r_i - r_b) = 2r_i(1 - q)$. In this case, as it is shown in Bruno (1991), the support of $\mu$ in (5) lies in an interval $[s_m, s_M]$, $0 < s_m < s_M < 1$ such that $s_m = \frac{1}{2}(1 - q^2), s_M = \frac{1}{2}(1 + q^2)$. The further the separation of the inclusions, the smaller the interval $[s_m, s_M]$, and the tighter the bounds. Smaller $q$ values indicate well separated brine (and colder temperatures as in Fig. 6), and $q = 1$ corresponds to no restriction on the separation, with $s_m = 0$ and $s_M = 1$.

Two parameters characterizing the structure of the sea ice composite are volume fraction $p$ of the brine inclusions and a separation parameter $q$ that quantifies how close the inclusions are to each other. Using observed values of effective complex permittivity, and inverting the forward matrix particle bounds, information about these two parameters is obtained in Orum et al. (2012) by solving exactly a reduced inverse spectral problem and bounding the volume fraction of the constituents, an inclusion separation parameter and the spectral gap of a self-adjoint operator that depends on the geometry of the composite.
Inverse bounds for inclusion separation are shown in Fig. 6 Orum et al. (2012), where the lower bound $q_{min}$ is displayed versus temperature of the sea ice slab. The inverted data clearly indicate that as the ice warms, the separations of the brine inclusions decrease. It is remarkable that this important phenomenon is characterized from electromagnetic measurements through an inversion scheme.

## 5.3 Extension to polycrystalline composites

The method of inverse bounds Cherkaev and Tripp (1996); Cherkaev and Golden (1998); Tripp et al. (1998) for structural parameters of a composite from measured effective properties was extended to polycrystalline materials in Gully et al. (2015). In the case of uniaxial polycrystalline composite, Gully et al. (2015) develops bounds for the mean orientation of crystals in the sea ice from measured values of ice permittivity. As columnar and granular microstructures have different mean single crystal orientations Weeks and Ackley (1982) this inverse approach is useful for determining ice type when using remote sensing
techniques.

The structures of different types of ice formed under different environmental conditions vary tremendously. For instance, for congelation ice frozen under calm conditions, the crystals are vertically elongated columns, and each crystal itself is a composite of pure ice platelets separating layers brine inclusions. The orientation of each crystal is determined by the direction that the c-axis points, which is perpendicular to the platelets. Finding the bounds for the crystals orientation we can electromagnetically





distinguish columnar ice from granular ice. This is a critical problem in sea ice physics and biology, as these different structures have vastly different fluid flow properties (with 5% *vs.* 10% brine volume fraction at the percolation threshold) which affects melt pond evolution, nutrient replenishment, brine convection, and other mesoscale processes in the ice cover.

Bounds for the effective permittivity of polycrystalline composites are much tighter than those bounding the permittivity of a general two-component material and statistically isotropic two-component material for sea ice. Such polycrystalline bounds

constructed in Gully et al. (2015) are shown in two right panes of Fig. 6(c). Polycrystalline bounds for the permittivity sea ice (left) Gully et al. (2015) (with the measured data on permittivity of sea ice Arcone et al. (1986)) provide a much tighter bound than general two-component material and statistically isotropic two-component material for sea ice given on the right (notice a different scale). This dramatic improvement over the classic two-component bounds is due to additional information about single crystal orientations included in the new bounds.

As was discussed in the polycrystal section, the zero moment $\mu_{kk}^0$ in (16) of the measure $\mu$ in the integral representation of the effective properties of a uniaxial polycrystalline material is $\mu_{kk}^0 = \langle |X_1 e_k|^2 \rangle$. The statistical average $\langle |X_1 e_k|^2 \rangle$ can be viewed as the "mean crystal orientation" related to the percentage of crystallites oriented in the $k^{\text{th}}$ direction.

Extending the inverse bounds method Cherkaev and Tripp (1996); Cherkaev and Golden (1998); Tripp et al. (1998) to polycrystalline materials, the inverse polycrystalline bounds Gully et al. (2015) estimate the mean crystal orientation by bounding

the zero moment $\mu_{kk}^0$ of the measure $\mu$ using measured data on the ice permittivity. This procedure gives an analytic estimate (the first order inverse bounds) for the range of values of the mean crystal orientation similar to (23):

$$\langle e_k^T X_1 e_k \rangle_l \leq \langle e_k^T X_1 e_k \rangle \leq \langle e_k^T X_1 e_k \rangle_u \,,$$
$$\langle e_k^T X_1 e_k \rangle_l = |f|^2 \frac{Im(\bar{s})}{Im(f)} \,, \qquad \langle e_k^T X_1 e_k \rangle_u = 1 - |g|^2 \frac{Im(\bar{t})}{Im(g)} \,, \tag{25}$$

Here $X_1$ is defined in the polycrystalline section as $X_1 = R^T C R$, $f$ is the known value of $F(s)$ and $g$ is the known value of

$G(t) = 1 - \epsilon^*/\epsilon_1$ with $t = 1 - s$.

Inverse polycrystalline bounds computed in Gully et al. (2015) for different types of sea ice, granular and columnar ice, show that the method allows revealing the type of ice based on electromagnetic data. For statistically isotropic granular ice shown in Fig. 5(a)-top, the inverse mean crystal orientation bounds Gully et al. (2015) estimate the deviation angle as $\pi/2 \pm .02$ (with the true value $\pi/2$). The inverse mean crystal orientation bounds Gully et al. (2015) for columnar ice (see Fig. 5(a)-bottom),

estimate the angle of deviation of the crystal's axis from the vertical as $20^o \pm 8^o$. These results demonstrate a significant difference in the reconstructed mean orientations of crystals in columnar and in granular ice and provide a foundation for distinguishing the types of ice using electromagnetic measurements.

Generalization of these polycrystalline bounds to the case when $c$-axis has a Gaussian distribution with known mean angle and the variance in the horizontal plane is developed in McLean et al. (2022) as a method for obtaining bounds on effective

permittivity of columnar sea ice that has a preferred direction in the horizontal plane due to a prevailing ocean current.



## 6 Analytic continuation for advection diffusion processes.

The enhancement of diffusive transport of passive scalars by complex fluid flow plays a key role in many important processes in the global climate system Washington and Parkinson (1986) and Earth's ecosystems Di Lorenzo et al. (2013). Advection of geophysical fluids intensifies the dispersion and large scale transport of heat Moffatt (1983), pollutants Csanady (1963); Beychok (1994); Samson (1988), and nutrients Di Lorenzo et al. (2013); Hofmann and Murphy (2004) diffusing in their environment. In sea ice dynamics, where the ice cover couples the atmosphere to the polar oceans Washington and Parkinson (1986), the transport of sea ice can also be enhanced by eddy fluxes and large scale coherent structures in the ocean Watanabe and Hasumi (2009); Lukovich et al. (2015); Dinh et al. (2022). In sea ice thermodynamics, the temperature field of the atmosphere is coupled to the temperature field of the ocean through sea ice, a composite of pure ice with brine inclusions whose volume fraction and connectedness depend strongly on temperature Thomas and Dieckmann (2003); Golden et al. (2007); Golden (2009). Convective brine flow through the porous microstructure can enhance thermal transport through the sea ice layer Lytle and Ackley (1996); Worster and Jones (2015); Kraitzman et al..

Over the years a broad range of mathematical techniques have been developed that reduce the analysis of complex composite materials, with rapidly varying structures in space, to solving averaged, or *homogenized* equations that do not have rapidly varying data, and involve an effective parameter. Motivated by Papanicolaou and Varadhan (1982), the effective parameter problem was extended to complex velocity fields, with rapidly varying structures in both space and time, yielding the effective (eddy) viscosity and the effective (eddy) diffusivity tensors McLaughlin et al. (1985). The effective parameter problem of (anomalous) super–diffusion and sub–diffusion is given in Biferale et al. (1995); Fannjiang (2000). Based on McLaughlin et al. (1985), Avellaneda and Majda Avellaneda and Majda (1989, 1991) adapted the ACM Golden and Papanicolaou (1983) to the advection diffusion equation and obtained a *Stieltjes integral representation* of the effective diffusivity tensor D$^*$, for flows with zero mean drift, involving the Péclet number $\xi$ of the flow. This representation encapsulates the geometric complexity of the flow in a spectral measure associated with a random Hermitian operator (or matrix). Mimicking methods developed for composite media Milton (2002), they obtained rigorous bounds on the components of D$^*$. Moreover, in direct analogue of methods developed for composites Milton (2002), they also found velocity fields which realize these bounds, such as the famous confocal sphere configurations which realize the Hashin–Shtrikman bounds of composites Hashin and Shtrikman (1962); Avellaneda and Majda (1991). Remarkably, this method has also been extended to time dependent flows Avellaneda and Vergassola (1995), flows with incompressible *nonzero* effective drift Pavliotis (2002); Fannjiang and Papanicolaou (1994), flows where particles diffuse according to linear collisions Pavliotis (2010), and solute transport in porous media Bhattacharya (1999), which has a direct application to diffusive brine advection in sea ice. All yield Stieltjes integral representations of the symmetric and, when appropriate, the antisymmetric part of D$^*$.

We now briefly describe our recent results on this framework Kraitzman et al.; Murphy et al. (2017b, 2020). It is an important example of how Stieltjes integral representations can provide a rigorous basis for analysis of problems for sea ice involving advection diffusion processes. The dispersion of a cloud of passive scalars with density $\phi(t, x)$ diffusing with molecular diffusivity $\varepsilon$ and being advected by a incompressible velocity field $u(t, x)$ satisfying $\nabla \cdot u = 0$ is described by the advection-diffusion





equation

$$\frac{\partial \phi}{\partial t} = u \cdot \nabla \phi + \epsilon \Delta \phi, \qquad \phi(0, x) = \phi_0(x). \tag{26}$$

Here, the initial density $\phi_0(x)$ and the fluid velocity field $u$ are assumed to be given. In equation (26), the molecular diffusion constant $\varepsilon > 0$, $d$ is the spatial dimension of the system, $\partial_t$ denotes partial differentiation with respect to time $t$, and $\Delta = \nabla \cdot \nabla = \nabla^2$ is the Laplacian. Moreover, $\boldsymbol{\psi} \cdot \boldsymbol{\varphi} = \boldsymbol{\psi}^T \overline{\boldsymbol{\varphi}}$, $\boldsymbol{\psi}^T$ denotes transposition of the vector $\boldsymbol{\psi}$, and $\overline{\boldsymbol{\varphi}}$ denotes component-

wise complex conjugation, with $\boldsymbol{\psi} \cdot \boldsymbol{\psi} = |\boldsymbol{\psi}|^2$. Later, we will use this form of the dot product over complex fields, with built in complex conjugation. However, we emphasize that all quantities considered in this section are *real-valued*. The random paths of the tracer particles are determined Fannjiang and Papanicolaou (1997) by the stochastic differential equation

$$\mathrm{d}x(t) = u(t, x(t))\mathrm{d}t + \sqrt{2\varepsilon}\,\mathrm{d}W(t), \quad x(0) = x_0, \tag{27}$$

with the initial tracked tracer particle location $x_0$ given and $W(t)$ is standard Brownian motion (the Wiener process). Non-

dimensionalizing and homogenizing (26) shows McLaughlin et al. (1985) that the effective behavior of thermal transport in sea ice is described by a diffusion equation involving an averaged scalar density $\bar{\phi}$ and a symmetric, constant Pavliotis (2002) effective diffusivity tensor $\boldsymbol{\kappa}^*$ Taylor (1921),

$$\frac{\partial \bar{T}(t, x)}{\partial t} = \nabla \cdot [\boldsymbol{\kappa}^* \nabla \bar{T}(t, x)], \quad \bar{T}(0, x) = T_0(x). \tag{28}$$

For simplicity, we focus on a diagonal coefficient $\kappa_{kk}^*$, $k = 1, \dots, d$, of $\boldsymbol{\kappa}^*$, set $\kappa^* = (\boldsymbol{\kappa}^*)_{kk}$, and write $u = u_0 v$ involving the

non-dimensional velocity field $v$. In these non-dimensional variables the Péclet number $\xi$ and molecular diffusivity $\varepsilon$ are related by $\xi = 1/\varepsilon$ Murphy et al. (2017b).

Using a mathematical framework that is strikingly similar to that in Section 3, the effective diffusivity has the following Stieltjes integral representation McLaughlin et al. (1985); Avellaneda and Majda (1991); Murphy et al. (2017b, 2020)

$$\kappa^* = \varepsilon(1 + \langle |\nabla w_k|^2 \rangle), \qquad \langle |\nabla w_k|^2 \rangle = \int_{-\infty}^{\infty} \frac{\mathrm{d}\nu(\lambda)}{\varepsilon^2 + \lambda^2}, \tag{29}$$

where $\langle \cdot \rangle$ denotes averaging over the space-time period cell for periodic flows Murphy et al. (2017b, 2020) or statistical average for random flows Avellaneda and Majda (1989); Avellaneda and Vergassola (1995). An equivalent statement which emphasizes the connection to the two component composites setting in equation (5) is

$$F(\varepsilon) = 1 - \frac{\kappa^*}{\varepsilon} = \int_{-\infty}^{\infty} \frac{\mathrm{d}\nu(\lambda)}{\varepsilon^2 + \lambda^2}. \tag{30}$$

Remarkably, the vector field $E(t, x) = \nabla w_k(t, x) + e_k$ satisfies equation (3) for two-component composite materials, with

$D = \boldsymbol{\epsilon} E$, $\boldsymbol{\epsilon} = \varepsilon I + S$, $S = (-\Delta)^{-1}\partial_t + H$, and $\boldsymbol{\epsilon}$ plays the role of the medium's electrical permittivity tensor Murphy et al. (2017b, 2020). Here, $H(t, x)$ is the *stream matrix*, given in terms of the incompressible velocity field $v = \nabla \cdot H$ and satisfies $H^T = -H$ Avellaneda and Majda (1991, 1989). When the flow is time-independent, $v = v(x)$, then $w_k = w_k(x)$ and $S =$





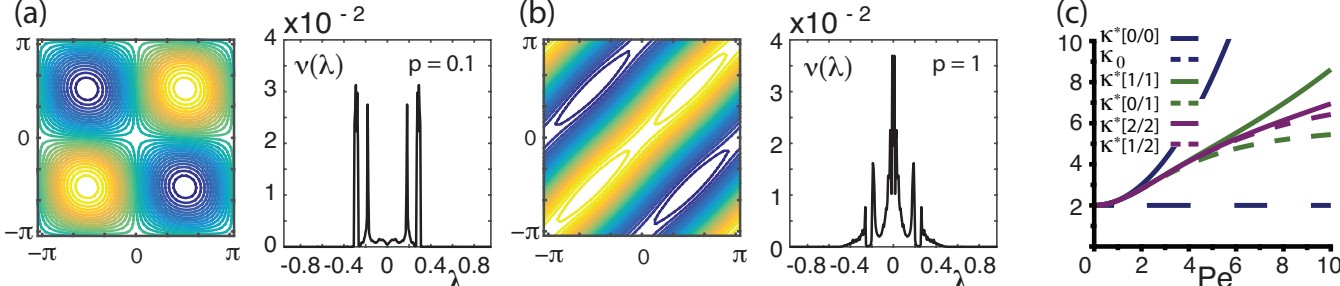

**Figure 7. Spectral behavior of homogenized diffusivities.** (a) Streamlines for BC-flow with velocity field $v = (C \cos y, B \cos x)$ and $B = C = 1$. (b) Padé approximant upper $\kappa^*[M/M]$ and lower $\kappa^*[M-1/M]$ bounds for $\kappa^*$, for various values of $M$, calculated for BC-flow with $C = B$, as a function of the flow strength $B$. (c) The spectral function (spectral masses $m_j$ versus eigenvalues $\lambda_j$) computed via analogues of equations (9) and (18) Murphy et al. (2017b).

$H(x)$. Moreover $\kappa^* = \epsilon^*$, with $\epsilon^* = (\epsilon^*)_{kk}$ defined above Murphy et al. (2017b). The integral representation for $\kappa^*$ in Equation (29) follows from the resolvent formula

$$\nabla w_k = (\varepsilon I + i\Gamma S\Gamma)^{-1} g_k, \quad g_k = -\Gamma H e_k \tag{31}$$

which is an analogue of Equation (6). The *self-adjoint* operator $i\Gamma S\Gamma$, where $i = \sqrt{-1}$ is the imaginary unit, involves the same projection operator $\Gamma = -\nabla(-\Delta)^{-1}\nabla\cdot$ as the setting of two-component composites. Equation (29) shows that brine advection *enhances* the thermal diffusivity (and the thermal conductivity) of sea ice, since $\kappa^* \geq \varepsilon$.

Analytical calculations of the spectral measure $\nu$ are extremely difficult except for simple flows like shear flow Avellaneda

and Majda (1991). However, Padé approximants $[L/M]$ provide rigorous, converging upper and lower bounds Baker and Graves-Morris (1996) for the *Stieltjes function* $f(z) = \langle|\nabla w_k|^2\rangle/z = F/z$ in Equations (29) and (30), with $z = \varepsilon^{-2}$, using the moments $\nu_n$ of $\nu$, $[M-1/M] \leq f(z) \leq [M/M]$, $f(z) = \sum_{n=0}^{\infty}(-1)^n \nu_{2n} z^n$. However, the lack of a method to calculate the moments $\nu_n$ of $\nu$ has impeded progress on obtaining explicit bounds for specific flows using this procedure Avellaneda and Majda (1991, 1989) since 1991! We have recently developed a mathematical framework Murphy et al. (2022) that can be used

to compute, in principle, *all* of the moments $\nu_n$ associated with a spatially or space-time periodic brine velocity field $v$, hence Padé approximant bounds. Results for BC-flow, with $v = (C \cos y, B \cos x)$ and $B = C$ are shown in Figure 7(c).

### 6.1 Spectral measure computations for advection diffusion processes.

We have extended our numerical methods discussed for the two component media to compute the spectral measure $\nu$ for spatially periodic flows Murphy et al. (2017b). Computing the spectral measure $\mu$ for a given flow involves discretizing the

spatially dependent stream matrix $H(x)$, which becomes a banded antisymmetric matrix satisfying $H^T = -H$. The projection matrix $\Gamma$ is given by that in Section 3.1 and the key self-adjoint operator is given by $G = i\Gamma H\Gamma$, which becomes a *Hermitian* matrix. In this case, the integral in (29) and the resolvent in (31) are given in terms of the eigenvalues and eigenvectors of the





matrix

$$\nabla w_k = \sum_i \frac{\sqrt{m_i}}{\varepsilon - \lambda_i} u_i\,, \quad \langle |\nabla w_k|^2 \rangle = \sum_i \left\langle \frac{m_i}{\varepsilon^2 + \lambda_i^2} \right\rangle\,, \quad m_i = |u_i \cdot g_i|^2\,, \tag{32}$$

which is analogous to equation (9). We have also developed Fourier methods for computing the spectral measure $\nu$ for space-time periodic flows Murphy et al. (2017b).

These computations show that the origin in the space of the spectral parameter $\lambda$ for advection diffusion plays the role of the spectral endpoints 0 and 1 for composite materials, with an increase in spectral mass giving rise to an advection-driven enhancement of effective diffusivity above the bare molecular diffusivity $\varepsilon$. For example, the closed streamlines of BC-cell-655 flow with fluid velocity field $v = (C\cos y, B\cos x)$ and $B = C = 1$ transport tracers in a short range periodic motion so long range transport is only possible due to molecular diffusion. Consequently, in the advection dominated regime with $\varepsilon \ll 1$ (or Péclet number $\xi \gg 1$) the effective diffusivity scales as $\kappa^* \sim \varepsilon^{1/2}$ Fannjiang and Papanicolaou (1994, 1997); Murphy et al. (2020), vanishing as $\varepsilon \to 0$. As shown in Fig. 7(a), this is reflected in the spectral measure $\nu$ by the lack of adequate mass near $\lambda = 0$ for the singular integrand $1/(\varepsilon^2 + \lambda^2)$ to overcome the multiplicative factor of $\varepsilon$ for $\kappa^* = \varepsilon(1 + \langle |\nabla w_k|^2 \rangle)$ in (29).

On the other hand, when $B \neq C$ the streamlines elongate and connect to neighboring cells which gives rise to long range advection of tracers, even in the absence of molecular diffusion. This is reflected in the spectral measure by a buildup of adequate mass near $\lambda = 0$ for the singular integrand $1/(\varepsilon^2 + \lambda^2)$ to overcome the multiplicative factor of $\varepsilon$ for $\kappa^* = \varepsilon(1 + \langle |\nabla w_k|^2 \rangle)$ in (29), leading to a non-zero value of $\kappa^*$ in the limit $\varepsilon \to 0$. This is a key example of how the behavior of the spectral measure $\nu$ governs the behavior of the bulk transport coefficient $\kappa^*$.

## 7 Random matrix theory for sea ice physics.

In random matrix theory (RMT) Guhr et al. (1998); Bohigas and Giannoni (1984); Deift and Gioev (2009), long and short range correlations of the bulk eigenvalues away from the spectral edge Canali (1996); Guhr et al. (1998) for random matrices are measured using various eigenvalue statistics Guhr et al. (1998); Bohigas and Giannoni (1984), such as the eigenvalue spacing distribution (ESD) and the spectral rigidity $\Delta_3$ and number variance $\Sigma^2$. To observe statistical fluctuations of these 670 bulk eigenvalues about the mean density, the eigenvalues must be unfolded Bohigas and Giannoni (1984); Guhr et al. (1998); Canali (1996); Plerou et al. (2002). The localization properties of the eigenvectors are measured in terms of quantities such as the inverse participation ratio (IPR) Plerou et al. (2002); Evers and Mirlin (2008).

In Murphy et al. (2017a), we found that as a percolation threshold is approached and long range order develops, the behavior of the ESD transitions from uncorrelated Poissonian toward obeying universal Wigner-Dyson (WD) statistics of the Gaussian 675 Orthogonal Ensemble (GOE). The eigenvectors de-localize, and mobility edges appear Murphy et al. (2017a), similar to the metal/insulator transition in solid state physics. We explored the transition in the 2D and 3D RRN, as well as in sea ice microstructures such as in 2D discretizations of the brine microstructure of sea ice Golden et al. (1998a, 2007); Golden (2009), melt ponds on Arctic sea ice Hohenegger et al. (2012), the sea ice pack itself, and porous human bone Golden et al. (2011); Kabel et al. (1999); Bonifasi-Lista and Cherkaev (2009); Cherkaev and Bonifasi-Lista (2011).



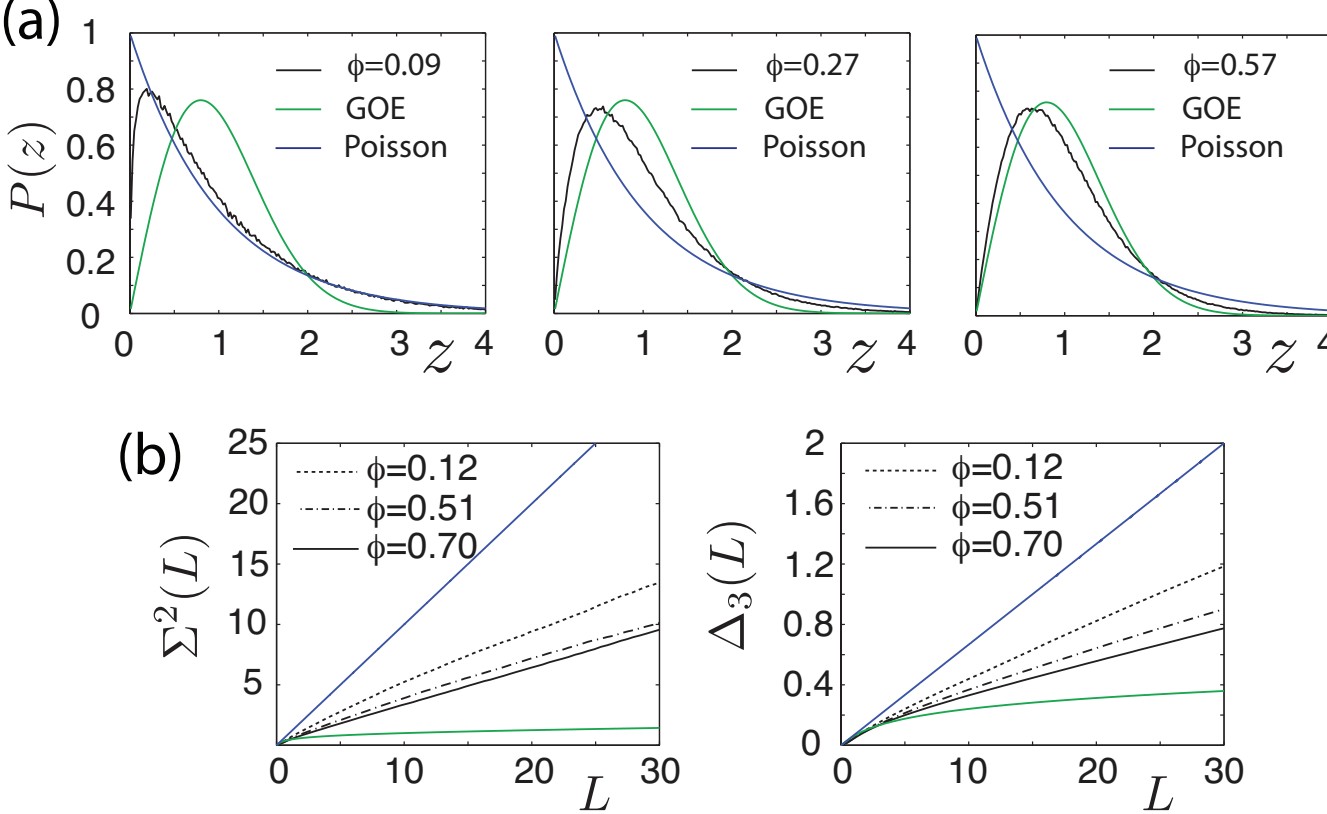

**Figure 8.** *Eigenvalue spacing statistics for the sea ice melt ponds (a) and long range eigenvalue statistics for brine structures in sea ice (b).* (a): Eigenvalue spacing distributions (ESD) $P(z)$ for melt ponds shown in Figure 4 corresponding to melt water area fractions 9%, 27%, and 57%. (b): Spectral statistics for brine structures shown in Figure 3 corresponding to area fractions of water 12%, 51%, and 70%. We see the transition to universal Wigner-Dyson statistics as ocean phases and brine phases become connected over the scale of the sample.

For highly correlated WD spectra exhibited by, for example, real-symmetric matrices of the GOE, the nearest neighbor ESD $P(z)$ is accurately approximated by $P(z) \approx (\pi z/2)\exp(-\pi z^2/2)$, which illustrates *eigenvalue repulsion*, vanishing linearly as spacings $z \to 0$ Guhr et al. (1998); Stone et al. (1991); Canali (1996). In contrast, the ESD for uncorrelated Poisson spectra, $P(z) = \exp(-z)$, allows for significant level degeneracy Guhr et al. (1998). In Fig. 8(a) we display the ESDs for Poisson (blue) and WD (green) spectra, along with the behavior of the ESDs for the matrix $M = \chi_1 \Gamma \chi_1$, corresponding to the arctic

sea ice melt ponds in Fig. 4 with fluid area fraction $\phi$. It shows that for sparsely connected systems, the behavior of the ESDs is well described by weakly correlated Poisson-like statistics Canali (1996). With increasing connectedness, the ESDs transition toward highly correlated WD statistics with strong level repulsion. This behavior of the ESD reveals a mechanism for the collapse in the spectral gaps of $\mu$. For sparsely connected systems, the weak level repulsion allows for significant level degeneracy and resonances in $\mu$ as shown in Murphy et al. (2015) for the 2D percolation model an in Fig. 4 for arctic melt pond

microstructure. As the system becomes increasingly connected, the level repulsion increases causing the eigenvalues to spread

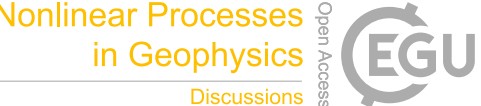

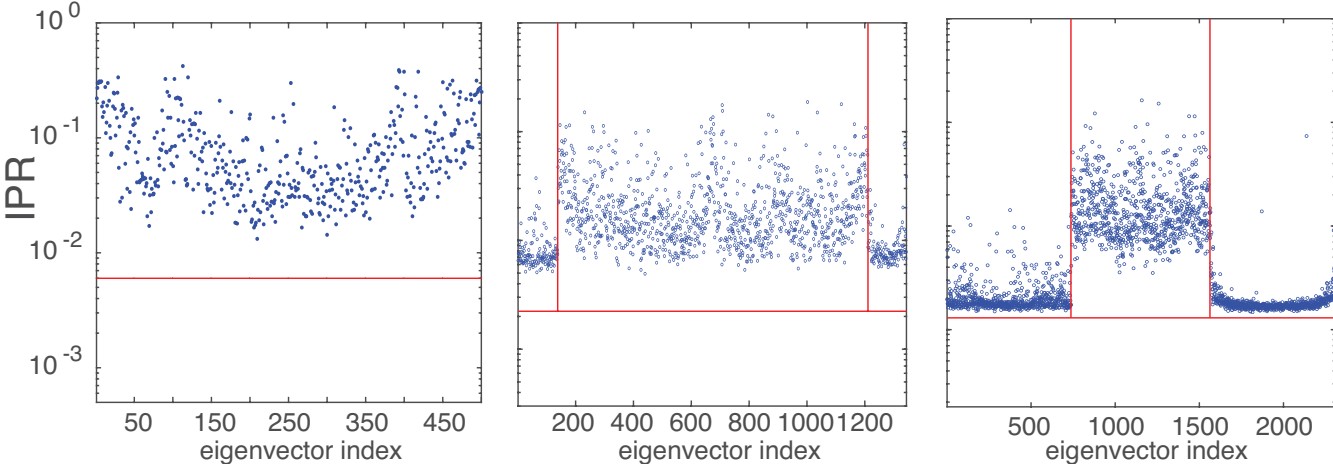

**Figure 9.** *Eigenvector localization for Arctic melt pond microstructure.* The IPR's of eigenvectors $u_j$ associated with the melt pond microstructure shown above plotted versus eigenvalue index $j$ and increasing connectedness from left to right. The vertical lines define the $\delta$-components of $\mu$ while the horizontal lines mark the IPR value $I_{GOE} = 3/N_1$ for the Gaussian orthogonal ensemble (GOE) with matrix size $N_1 \approx \phi N$, where $N = \mathrm{L}^d d$.

out which, in turn, causes the gaps in the measure near the spectral edges to collapse and subsequently form $\delta$-components of the measure at the spectral endpoints $\lambda = 0, 1$. Our computations of $\Delta_3$ and $\Sigma^2$ are are shown in Fig. 8(b) for the brine microstructure in Fig. 3, with a transition toward that of the GOE, as the system becomes increasingly connected, indicating an increase in the long range correlations of the eigenvalues.

The eigenvectors $u_j$ of $M = \chi_1 \Gamma \chi_1$, associated with the $N_1 \times N_1$ sub-matrices of $\Gamma$, also exhibit a connectedness driven transition in their localization properties. The IPR is defined as $I_j = \sum_i |u_j^i|^4$, $i, j = 1, \ldots, N_1$, where $u_j^i$ is the $i$th component of $u_j$. Eigenvectors of matrices in the GOE are known to be highly extended Deift and Gioev (2009), with asymptotic value of the IPR given by $I_{GOE} = 3/N_1$ Plerou et al. (2002). In Murphy et al. (2017a), we found for the 2D and 3D percolation models that as $p$ surpasses $p_c$ and long range order is established in a RRN **"mobility edges"** form in the eigenvector IPR with a sudden

increase in the number of extended eigenvectors, which is analogous to Anderson localization, where mobility edges mark the characteristic energies of the MIT Guhr et al. (1998). Remarkably, the mobility edges for RRN are due to very extended eigenstates associated with $\delta$-components that form at the spectral endpoints *precisely* at the percolation threshold $p_c$ (and $1 - p_c$ for 3D) Murphy and Golden (2012), which control critical behavior in insulator/conductor and conductor/superconductor systems Murphy and Golden (2012); Clerc et al. (1990); Bergman and Stroud (1992). This phenomenon is shown for arctic

melt pond microstructure in Fig. 9.





## 8    Conclusions

We have given a tour through various problems of sea ice physics concerned with homogenization and how they can be rigorously addressed with the powerful analytic continuation method and its extensions. The effective complex permittivity of sea ice treated as a two phase composite of pure ice with brine inclusions, or treated as a polycrystalline material, and

the effective diffusivity for advection diffusion problems, are all Stieltjes functions of their variables. We showed how these functions have integral representations involving spectral measures which distill the mixture or velocity field geometries into the spectral properties of a self adjoint opeartor like the Hamiltonian in quantum physics. These spectral representations have been used to obtain rigorous forward and inverse bounds on effective transport coefficients for sea ice, and to develop a random matrix theory picture which uncovers parallels with Anderson localization and quantum transport in disordered media.

**Appendix A:  Existence of field decompositions**

In this section, for the discrete setting in Sections 3.1 and 4.1, we prove that there exists an electric field $E$ satisfying discrete versions of equations (3) and (4). Towards this goal, we follow Huang et al. (2019) and consider the finite difference representations of the partial differential operators $\partial_i \to C_i$, $i = 1 \dots, d$, where $d$ denotes dimension. The matrices $C_i$ depend on boundary conditions which, without loss of generality, we take to be periodic boundary conditions. Denote the matrix represen-

tation of the gradient operator (using Matlab vertical block notation) by $\nabla = [C_1; \dots; C_d]$. The discretization of the divergence operator is given by $-\nabla^T$ and the discrete curl operator is given by Huang et al. (2019)

$$C = \begin{bmatrix} O & -C_3 & C_2 \\ C_3 & O & -C_1 \\ -C_2 & C_1 & O \end{bmatrix} \text{ for 3D,} \tag{A1}$$

$C = [-C_2, C_1]$ for 2D.

The operators $C_i$, $i = 1, 2, 3$, in (A8) are normal and commute with each other Huang et al. (2019),

$C_i^T C_j = C_j C_i^T$ and $C_i C_j = C_j C_i$, for $i, j = 1, 2, 3$. $\tag{A2}$

Consequently, the discrete form of equations (3) and (4), which we'll establish in this section, is

$$CE = 0, \quad -\nabla^T J = 0, \quad J = \epsilon E, \quad E = E_f + E_0, \quad CE_f = 0, \quad \langle J \cdot E_f \rangle = 0, \quad \langle E_f \rangle = 0, \tag{A3}$$

where in this finite size discrete setting, $\langle \cdot \rangle$ denotes volume average followed by ensemble average Murphy et al. (2015, 2020). To set notation, denote by $\mathrm{Ran}(B)$ and $\mathrm{Ker}(B)$ the range and kernel (null space) of the matrix $B$, respectively Horn and Johnson

(1990). Therefore, we seek to prove there exists a vector $E$ satisfying $E \in \mathrm{Ker}(C)$ such that $E = E_f + E_0$, where $E_f \in \mathrm{Ker}(C)$ and $\langle E_f \rangle = 0$ so $\langle E \rangle = E_0$. Moreover, we seek to find a vector $J \in \mathrm{Ker}(\nabla^T)$ satisfying $J = \epsilon E$ and $\langle J \cdot E_f \rangle = 0$.





We now summarize some useful identities relating the discrete representations of the gradient, divergence, and curl operators which follow from these properties of the matrices $C_i$ Huang et al. (2019),

$$\Delta = \nabla \cdot \nabla \to -\nabla^T \nabla\,, \tag{A4}$$

$$\boldsymbol{\Delta} = \mathrm{diag}(\Delta,\dots,\Delta) \to I_d \otimes (\nabla^T \nabla)\,,$$

$$\nabla \times \nabla \times \to C^T C\,,$$

$$\nabla \times \nabla \times = \nabla(\nabla \cdot) - \boldsymbol{\Delta} \to -\nabla\nabla^T + I_d \otimes (\nabla^T \nabla)\,,$$

$$\nabla \cdot (\nabla \times) \to -\nabla^T C^T = 0\,,$$

$$\nabla \times \nabla \to C\nabla = 0\,,$$

where $\otimes$ denotes the Kronecker product. The last two identities $\nabla^T C^T = 0$ and $C\nabla = 0$ in equation (A4) indicate that

$$\mathrm{Ran}(C^T) \subseteq \mathrm{Ker}(\nabla^T)\,, \quad \mathrm{Ran}(\nabla) \subseteq \mathrm{Ker}(C) \tag{A5}$$

We can now restate our goal in (A3) as, find "potentials" $\varphi$ and $\psi$ such that the vectors $E$ and $J$ in equation (A3) satisfy $E = E_f + E_0$ with $E_f = \nabla\varphi$ and $E_0 \in \mathrm{Ker}(\nabla)$, and $J = C^T \psi + J_0$, where $J_0 \in \mathrm{Ker}(C^T)$. The last two identities (A4) provide a relationship between rank and kernel of the operators $C$, $\nabla$, and their transposes. The fundamental theorem of linear algebra provides a relationship between the range of a matrix $A$ and the kernel of it's transpose $A^T$, which will be useful later in this section.

**Theorem 3** (Fundamental theorem of linear algebra). *Let $A$ be a real valued matrix of size $m \times n$ then*

$$\mathbb{R}^m = \mathrm{Ran}(A) \oplus \mathrm{Ker}(A^T)\,, \qquad \mathbb{R}^n = \mathrm{Ran}(A^T) \oplus \mathrm{Ker}(A)\,, \tag{A6}$$

*where $\oplus$ indicates $\mathrm{Ran}(A)$ is orthogonal to $\mathrm{Ker}(A^T)$, i.e., $\mathrm{Ran}(A) \perp \mathrm{Ker}(A^T)$, for example.*

Applying Theorem 3 to the matrices $\nabla$ and $C^T$ indicates that $\mathbb{R}^m = \mathrm{Ran}(\nabla) \oplus \mathrm{Ker}(\nabla^T)$ and $\mathbb{R}^m = \mathrm{Ran}(C^T) \oplus \mathrm{Ker}(C)$. Therefore, from equation (A5) we have that divergence-free fields are orthogonal to gradients (curl-free fields) and curl-free fields are orthogonal to $\mathrm{Ran}(C^T)$ (divergence-free fields). This is a discrete version of the Helmholz Theorem, which states that curl-free and divergence-free fields (or, in other words, the gradient and cycle spaces) are mutually orthogonal. This also establishes the important relationship

$$\mathrm{Ran}(C^T) \perp \mathrm{Ran}(\nabla)\,. \tag{A7}$$

Orthogonal bases can be given for each of the mutually orthogonal spaces in equation (A6) through the singular value decomposition (SVD) Horn and Johnson (1990) of the matrix $A = U\Sigma V^T$, which also provides important information relating the matrices $C$, $\nabla$, etc. Here $U$ and $V$ are orthogonal matrices of size $m \times m$ and $n \times n$ satisfying $U^T U = UU^T = I_m$ and $V^T V = VV^T = I_n$, where $I_m$ is the identity matrix of size $m$. Moreover, $\Sigma$ is a diagonal matrix of size $m \times n$ with diagonal



components consisting of the positive *singular values* $\nu_i$, $i = 1, \ldots, n$, of the matrix $A$ satisfying $\nu_1 \geq \nu_2 \geq \cdots \geq \nu_\rho > 0$ and $\nu_{\rho+1} = \nu_{\rho+2} = \cdots = \nu_n = 0$, where $\rho$ is the rank of $A$. Writing the matrix $\Sigma$ in block form we have

$$\Sigma = \begin{bmatrix} \Sigma_1 & O_1 \\ O_1^T & O_2 \\ O_3 & O_4 \end{bmatrix}, \tag{A8}$$

where $\Sigma_1$ is a diagonal matrix of size $\rho \times \rho$ with diagonal consisting of the strictly positive singular values, $O_1$ and $O_2$, are matrices of zeros of size $\rho \times (n - \rho)$ and $(n - \rho) \times (n - \rho)$, and the bottom block of zeros $[O_3, O_4]$ is of size $(m - n) \times n$.

Write the matrices $U$ and $V$ in block form as $U = [U_1, U_0, U_2]$ and $V = [V_1, V_0]$, where $U_1$ and $V_1$ are the columns of $U$ and $V$ corresponding to the strictly positive singular values in $\Sigma_1$, $U_0$ and $V_0$ correspond to the the singular values with value zero, $\nu_i = 0$, and $U_2$ corresponds to the bottom block of zeros $[O_3, O_4]$ in $\Sigma$. Taking in account all the blocks of zeros in $\Sigma$, we can write $A = U_1 \Sigma_1 V_1^T$. We now state a well known fact about the SVD of the matrix $A$ Horn and Johnson (1990).

$$\text{Range}(A) = \text{Col}(U_1), \quad \text{Ker}(A) = \text{Col}(V_0), \quad \text{Range}(A^T) = \text{Col}(V_1), \quad \text{Ker}(A^T) = \text{Col}([U_0, U_2]), \tag{A9}$$

where $\text{Col}(B)$ denotes the column space of the matrix $B$, i.e., the space spanned by the columns of $B$.

Applying the SVD to the matrices $\nabla = U^\times \Sigma^\times [V^\times]^T$ and $C^T = U^\bullet \Sigma^\bullet [V^\bullet]^T$ and using the orthogonality of the columns of the matrices $U_1$ and $V_1$ and the invertability of $\Sigma_1$, from $C\nabla = 0$ in (A4) we have $[U^\bullet]^T U_\times = 0$, and similarly $\nabla^T C^T = 0$ implies $[U^\times]^T U_\bullet = 0$. The formula $[U^\bullet]^T U_\times = 0$ is a restatement of equation (A7). Writing $U^\times = [U_1^\times, U_0^\times, U_2^\times]$ and $U^\bullet = [U_1^\bullet, U_0^\bullet, U_2^\bullet]$ we have established the $\text{Ran}(\nabla) = \text{Col}(U_1^\times)$, $\text{Ran}(C^T) \subseteq \text{Ker}(\nabla^T) = \text{Col}([U_0^\times, U_2^\times])$. Also, since $\text{Ran}(C^T) = $
$\text{Col}(U_1^\bullet)$ and $\text{Ran}(C^T) \perp \text{Ran}(\nabla)$, we can write

$$U^\times = [U_1^\times, U_0^{\times\bullet}, U_1^\bullet], \quad U^\bullet = [U_1^\bullet, U_0^{\times\bullet}, U_1^\times], \tag{A10}$$

where the columns of $U_0^{\times\bullet}$ are orthogonal to both the $\text{Ran}(C^T)$ and the $\text{Ran}(\nabla)$. Since $U^\times [U^\times]^T = I_m$ we have the *resolution of the identity*

$$\Gamma_\times + \Gamma_0 + \Gamma_\bullet = I_m, \qquad \Gamma_\times = U_1^\times [U_1^\times]^T, \quad \Gamma_\bullet = U_1^\bullet [U_1^\bullet]^T, \quad \Gamma_0 = U_0^{\times\bullet} [U_0^{\times\bullet}]^T, \tag{A11}$$

where $\Gamma_\times$, $\Gamma_\bullet$, and $\Gamma_0$, are mutually orthogonal projection matrices onto $\text{Ran}(\nabla)$, $\text{Ran}(C^T)$ and the orthogonal complement of $\text{Ran}(\nabla) \cup \text{Ran}(C^T)$. We are now ready to state the main result of this section as the following theorem.

**Theorem 4.** *Let the electric and current fields $E$ and $J$ satisfy*

$$CE = 0, \quad -\nabla^T J = 0, \quad J = \epsilon E. \tag{A12}$$

*Then, there exists a "potential" $\varphi$ and vector $E_0$ such that $E = E_f + E_0$, where $E_f = \nabla \varphi$, $I_d \otimes (\nabla^T \nabla)) E_0 = 0$, $CE_f = 0$,*
$\langle J \cdot E_f \rangle = 0$, *and $\langle E_f \rangle = 0$.*

*Proof.* From the resolution of the identity in equation (A11) we have $E = (\Gamma_\times + \Gamma_0 + \Gamma_\bullet) E$. Since $\Gamma_\bullet$ projects onto the $\text{Ran}(C^T)$, $\mathbb{R}^m = \text{Ran}(C^T) \oplus \text{Ker}(C)$, and $E \in \text{Ker}(C)$ we have $\Gamma_\bullet E = 0$. Denoting by $E_f = \Gamma_\times E$, since $U_1^\times = \nabla V_1^\times [\Sigma_1^\times]^{-1}$,





we can write $E_f = \nabla\varphi$, where $\varphi = V_1^\times[\Sigma_1^\times]^{-1}[U_1^\times]^T E$. Denote $E_0 = \Gamma_0 E$. Since $\Gamma_\times E_0 = 0$, $\Gamma_\times$ is a projection onto $\text{Ran}(\nabla)$, and $\mathbb{R}^m = \text{Ran}(\nabla) \oplus \text{Ker}(\nabla^T)$, we have $E_0 \in \text{Ker}(\nabla^T)$. Similarly, since $\Gamma_\bullet E_0 = 0$, $\Gamma_\bullet$ is a projection onto $\text{Ran}(C^T)$, and

$\mathbb{R}^m = \text{Ran}(C^T) \oplus \text{Ker}(C)$, we have $E_0 \in \text{Ker}(C)$. Since $E_0 \in \text{Ker}(C) \cap \text{Ker}(\nabla^T)$ we have from (A4) that

$$0 = C^T C E_0 = (-\nabla\nabla^T + I_d \otimes (\nabla^T\nabla))E_0 = I_d \otimes (\nabla^T\nabla))E_0. \tag{A13}$$

From $\text{Ran}(\nabla) \subseteq \text{Ker}(C)$ in equation (A5) and $E_f = \nabla\varphi$ we have $CE_f = C\nabla\varphi = 0$. We also have from $\nabla^T J = 0$ that $J \cdot E_f = J \cdot \nabla\varphi = \nabla^T J \cdot \varphi = 0$. Finally, the volume average of $\nabla\varphi$ is a telescoping sum, so $\langle E_f \rangle = 0$. This concludes our proof of the theorem. $\square$

**Corollary 1.** *Let the electric and current fields $E$ and $J$ satisfy*

$$CE = 0, \quad -\nabla^T J = 0, \quad J = \boldsymbol{\sigma} E. \tag{A14}$$

*Then, there exists a "potential" $\psi$ and vector $J_0$ such that $J = J_f + J_0$, where $J_f = C^T\psi$, $I_d \otimes (\nabla^T\nabla))J_0 = 0$, $\nabla^T J_f = 0$, $\langle J_f \cdot E \rangle = 0$, and $\langle J_f \rangle = 0$.*

*Proof.* From the resolution of the identity in equation (A11) we have $J = (\Gamma_\times + \Gamma_0 + \Gamma_\bullet)J$. Since $\Gamma_\times$ projects onto the $\text{Ran}(\nabla)$,

$\mathbb{R}^m = \text{Ran}(\nabla) \oplus \text{Ker}(\nabla^T)$, and $J \in \text{Ker}\nabla^T$ we have $\Gamma_\times J = 0$. Denoting by $J_f = \Gamma_\bullet J$, since $U_1^\bullet = C^T V_1^\bullet[\Sigma_1^\bullet]^{-1}$, we can write $J_f = C^T\psi$, where $\psi = V_1^\bullet[\Sigma_1^\bullet]^{-1}[U_1^\bullet]^T J$. Denote $J_0 = \Gamma_0 J$. Exactly the same as in the proof of Theorem 4, we have $J_0 = I_d \otimes (\nabla^T\nabla))J_0$. From $\text{Ran}(C^T) \subseteq \text{Ker}(\nabla^T)$ in equation (A5) and $J_f = C^T\psi$ we have $\nabla^T J_f = \nabla^T C^T\psi = 0$. Exactly the same as in the proof of Theorem 4, we also have $J_f \cdot E = 0$ and $\langle J_f \rangle = 0$. This concludes our proof of the theorem. $\square$

We end this section by noting that in the full rank setting, where $\Sigma$ has all strictly positive singular values, so $\Sigma_1 = \Sigma$, then

$\Gamma_\times = \nabla(\nabla^T\nabla)^{-1}\nabla^T, \qquad \Gamma_\bullet = C^T(CC^T)^{-1}C.$

The original formulations of this mathematical framework was given in terms of these projection matrices, or continuum generalizations Golden and Papanicolaou (1983); Murphy et al. (2015). The formulation given in this section generalizes the discrete setting to cases where the matrix gradient is rank deficient, such as the case of periodic boundary conditions. This is necessary

**Appendix B: Proof of Theorem 2**

In this section we provide the proof for Theorem 2 and a projection method for a numerically efficient projection method for computation of spectral measures and effective parameters for uniaxial polycrystalline media. We will use the results from Section A but for notational simplicity we will use $\Gamma$ instead of $\Gamma_\times$. Corollary 2 below follows immediately from the proof of Theorem 2 and the results of Section A, which provides an integral representation for the effective resistivity $\rho^*$ involving the

matrix $X_2\Gamma_\bullet X_2$ and is analogous to the representation of $\rho^*$ for the two-component composite setting in Murphy et al. (2015).





From the close analogues of this polycrystalline setting with the two-component setting discussed in Sec. 3, the proof of this theorem is analogous to Theorem 2.1 in Murphy et al. (2015). To shorten the theorem proof here, we will refer to Murphy et al. (2015) for some of the technical details. From Section A and the paragraph in Murphy et al. (2015) containing equations 2.39 and 2.40, with $\chi_1$ replaced by $X_1$, we just need to prove that the functional $F_{jk}(s) = \langle (sI - X_1 \Gamma X_1)^{-1} X_1 \hat{e}_j \cdot \hat{e}_k \rangle$ has the

integral representation displayed in equation (18). In the process, we will also establish a projection method for the numerically efficient, rigorous computation of $\mu_{jk}$. This projection method is summarized by equations (C1)–(C3) below.

In equation (14) we defined the real-symmetric mutually orthogonal projection matrices $X_i$, $i = 1, 2$, in terms of the *spatially dependent* rotation matrix $R$ and $C = \text{diag}(1, 0, \ldots, 0)$, all matrices of size $d \times d$. The paragraph in Murphy et al. (2015) containing equations 2.28–2.30 describes how to bijectively map these $d \times d$ matrices to $N \times N$ matrices that are *not* spatially

dependent, where $N = L^d d$. Under this mapping, $R$ becomes a banded rotation matrix satisfying $R^T = R^{-1}$ and $C$ becomes $C = \text{diag}(I_1, 0_1, \ldots, 0_1)$, where $I_1$ and $0_1$ are the identity and null matrices of size $N_1 = L^d$, and the vector $e_1$ is mapped to $\hat{e}_1 = (1, 1, \ldots, 1, 0, 0, \ldots, 0)$, with $L^d$ ones in the first components and zeros in the rest of the components. Writing $X_1 \Gamma X_1 = R^T (C R \Gamma R^T C) R$ we have

$$
\begin{aligned}
X_1 \Gamma X_1 &= R^T \begin{bmatrix} \Gamma_1 & 0_{12} \\ 0_{21} & 0_{22} \end{bmatrix} R = R^T \begin{bmatrix} W_1 \Lambda_1 W_1^T & 0_{12} \\ 0_{21} & 0_{22} \end{bmatrix} R \\
&= R^T \begin{bmatrix} W_1 & 0_{12} \\ 0_{21} & I_2 \end{bmatrix} \begin{bmatrix} \Lambda_1 & 0_{12} \\ 0_{21} & 0_{22} \end{bmatrix} \begin{bmatrix} W_1^T & 0_{12} \\ 0_{21} & I_2 \end{bmatrix} R,
\end{aligned}
\tag{B1}
$$

where $I_2$ is the identity matrix of size $N_2 \times N_2$, with $N_2 = N - N_1 = L^d(d-1)$, and $0_{ab}$ denotes a matrix of zeros of size $N_a \times N_b$, $a, b = 1, 2$. Moreover, $\Gamma_1$ is the $N_1 \times N_1$ upper left principal sub-matrix of $R \Gamma R^T$ and $\Gamma_1 = W_1 \Lambda_1 W_1^T$ is its eigenvalue decomposition. As $\Gamma_1$ is a real-symmetric matrix, $W_1$ is an orthogonal matrix Horn and Johnson (1990). Also, since $R \Gamma R^T$ is a similarity transformation of a projection matrix and $C$ is a projection matrix, $\Lambda_1$ is a diagonal matrix with entries $\lambda_i^1 \in [0, 1]$,

$i = 1, \ldots, N_1$, along its diagonal Horn and Johnson (1990); Demmel (1997).

Consequently, equation (B1) implies that the eigenvalue decomposition of the matrix $X_1 \Gamma X_1$ is given by

$$
X_1 \Gamma X_1 = W \Lambda W^T, \qquad W = R^T \begin{bmatrix} W_1 & 0_{12} \\ 0_{21} & I_2 \end{bmatrix}, \quad \Lambda = \begin{bmatrix} \Lambda_1 & 0_{12} \\ 0_{21} & 0_{22} \end{bmatrix}.
\tag{B2}
$$

Here, $W$ is an orthogonal matrix satisfying $W^T W = W W^T = I$, $I$ is the identity matrix on $\mathbb{R}^N$, and $\Lambda$ is a diagonal matrix with entries $\lambda_i \in [0, 1]$, $i = 1, \ldots, N$, along its diagonal.

The eigenvalue decomposition of the matrix $X_1 \Gamma X_1$ in equation (B2) demonstrates that its resolvent $(sI - X_1 \Gamma X_1)^{-1}$ is well defined for all $s \in \mathbb{C} \backslash [0, 1]$. In particular, by the orthogonality of the matrix $W$, it has the following useful representation $(sI - X_1 \Gamma X_1)^{-1} = W (sI - \Lambda)^{-1} W^T$, where $(sI - \Lambda)^{-1}$ is a diagonal matrix with entries $1/(s - \lambda_i)$ along its diagonal. This, in turn, implies that the functional $F_{jk}(s) = \langle (sI - X_1 \Gamma X_1)^{-1} X_1 \hat{e}_j \cdot \hat{e}_k \rangle$ can be written as

$$
F_{jk}(s) = \langle (sI - \Lambda)^{-1} [X_1 W]^T \hat{e}_j \cdot W^T \hat{e}_k \rangle.
\tag{B3}
$$





Since $R^T = R^{-1}$ and $X_1 = R^T C R$, equation (B2) implies that

$$X_1 W = R^T \begin{bmatrix} W_1 & 0_{12} \\ 0_{21} & 0_{22} \end{bmatrix} \quad \Longrightarrow \quad X_1 \mathrm{w}_i = \begin{cases} \mathrm{w}_i, & \text{for } i = 1, \dots, N_1, \\ 0, & \text{otherwise.} \end{cases} \tag{B4}$$

This and the formula for $W$ in (B2) imply that

$$[X_1 W]^T \hat{e}_j \cdot W^T \hat{e}_k = [X_1 W]^T \hat{e}_j \cdot [X_1 W]^T \hat{e}_k. \tag{B5}$$

We are ready to provide the integral representation displayed in (18) for the functional $F_{jk}(s)$ in equation (B3). Denote

by $Q_i = \mathrm{w}_i \mathrm{w}_i^T$, $i = 1, \dots, N$, the symmetric, mutually orthogonal projection matrices, $Q_\ell Q_m = Q_\ell \delta_{\ell m}$, onto the eigen-spaces spanned by the orthonormal eigenvectors $\mathrm{w}_i$. Equation (B4) implies that $X_1 Q_i = Q_i X_1 = X_1 Q_i X_1$, as $X_1 Q_i = Q_i$ for $i = 1, \dots, N_1$, $X_1 Q_i = 0$ otherwise, and $X_1$ is a symmetric matrix. These properties allow us to write the quadratic form $[X_1 W]^T \hat{e}_j \cdot [X_1 W]^T \hat{e}_k$ as

$$[X_1 W]^T \hat{e}_j \cdot [X_1 W]^T \hat{e}_k = W^T \hat{e}_j \cdot W^T \hat{e}_k = \sum_{i=1}^N (\mathrm{w}_i \cdot \hat{e}_j)(\mathrm{w}_i \cdot \hat{e}_k) = \sum_{i=1}^N Q_i \hat{e}_j \cdot \hat{e}_k = \sum_{i=1}^N X_1 Q_i \hat{e}_j \cdot \hat{e}_k. \tag{B6}$$

This and equations (B3) and (B5) yield

$$F_{jk}(s) = \int_0^1 \frac{\mathrm{d}\mu_{jk}(\lambda)}{s - \lambda}, \quad \mathrm{d}\mu_{jk}(\lambda) = \sum_{i=1}^N \langle \delta_{\lambda_i}(\mathrm{d}\lambda) \, X_1 Q_i \hat{e}_j \cdot \hat{e}_k \rangle. \tag{B7}$$

This concludes our proof of Theorem 2

**Corollary 2.** *For each $\omega \in \Omega$, let $M(\omega) = W(\omega) \Lambda(\omega) W(\omega)$ be the eigenvalue decomposition of the real-symmetric matrix $M(\omega) = X_2(\omega) \Gamma_\bullet X_2(\omega)$. Here, the columns of the matrix $W(\omega)$ consist of the orthonormal eigenvectors $\mathrm{w}_i(\omega)$, $i = 1, \dots, N$,*

*of $M(\omega)$ and the diagonal matrix $\Lambda(\omega) = \mathrm{diag}(\lambda_1(\omega), \dots, \lambda_N(\omega))$ involves its eigenvalues $\lambda_i(\omega)$. Denote $Q_i = \mathrm{w}_i \mathrm{w}_i^T$ the projection matrix onto the eigen-space spanned by $\mathrm{w}_i$. The curent field $J(\omega)$ satisfies $J(\omega) = J_0 + J_f(\omega)$, with $J_0 = \langle J(\omega) \rangle$ and $\Gamma_\bullet J(\omega) = J_f(\omega)$, and the effective complex resistivity tensor $\boldsymbol{\rho}^*$ has components $\rho_{jk}^*$, $j, k = 1, \dots, d$, which satisfy*

$$\rho_{jk}^* = \sigma_1^{-1}(\delta_{jk} - E_{jk}(s)), \qquad E_{jk}(s) = \int_0^1 \frac{\mathrm{d}\eta_{jk}(\lambda)}{s - \lambda}, \qquad \mathrm{d}\eta_{jk}(\lambda) = \sum_{i=1}^N \langle \delta_{\lambda_i}(\mathrm{d}\lambda) \, X_2 Q_i \hat{e}_j \cdot \hat{e}_k \rangle. \tag{B8}$$

**Appendix C: Projection method**

In this section we provide a formulation for a numerically efficient *projection method* for computation of spectral measures and effective parameters for uniaxial polycrystalline media. Note that the sum inequation (B7) runs only over the index set $i = 1, \dots, N_1$, as equation (B4) implies that the masses $X_1 Q_i \hat{e}_j \cdot \hat{e}_k$ of the measure $\mu_{jk}$ are zero for $i = N_1 + 1, \dots, N$. Denote by $\lambda_i^1$ and $\mathrm{w}_i^1$, $i = 1, \dots, N_1$, the eigenvalues and eigenvectors of the $N_1 \times N_1$ matrix $\Gamma_1 = W_1 \Lambda_1 W_1^T$, defined in equation (B1).



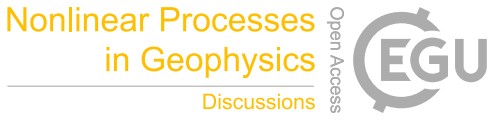
Now, write

$$R\hat{e}_j = \begin{bmatrix} \hat{e}_j^{r_1} \\ \hat{e}_j^{r_2} \end{bmatrix}, \tag{C1}$$

where $\hat{e}_j^{r_1} \in \mathbb{R}^{N_1}$ and $\hat{e}_j^{r_2} \in \mathbb{R}^{N_2}$. Therefore, writing the matrix $X_1 W$ in equation (B4) in block diagonal form, $X_1 W = R^T \mathrm{diag}(W_1, 0_{22})$, we have that

$$[X_1 W]^T \hat{e}_j \cdot [X_1 W]^T \hat{e}_k = [\mathrm{diag}(W_1^T, 0_{22}) R \hat{e}_j] \cdot [\mathrm{diag}(W_1^T, 0_{22}) R \hat{e}_k] = [W_1^T \hat{e}_j^{r_1}] \cdot [W_1^T \hat{e}_k^{r_1}]. \tag{C2}$$

Denote by $Q_i^1 = \mathrm{w}_i^1 [\mathrm{w}_i^1]^T$, $i = 1, \ldots, N_1$, the mutually orthogonal projection matrices, $Q_\ell^1 Q_m^1 = Q_\ell^1 \delta_{\ell m}$, onto the eigenspaces spanned by the orthonormal eigenvectors $\mathrm{w}_i^1$. Equations (B3), (B5), and (C2) then yield

$$F_{jk}(s) = \langle (sI_1 - \Lambda_1)^{-1} [W_1^T \hat{e}_j^{r_1}] \cdot [W_1^T \hat{e}_k^{r_1}] \rangle = \left\langle \sum_{i=1}^{N_1} \frac{Q_i^1 \hat{e}_j^{r_1} \cdot \hat{e}_k^{r_1}}{s - \lambda_i^1} \right\rangle. \tag{C3}$$

Equation (C3) demonstrates that only the spectral information of the matrices $W_1$ and $\Lambda_1$ contribute to the functional representation for $F_{jk}(s)$ in (B3) and its integral representation in (18). From a computational standpoint, this means that only the eigenvalues and eigenvectors of the $N_1 \times N_1$ matrix $\Gamma_1$ need to be computed in order to compute the spectral measures underlying the integral representations of the effective parameters for finite lattice systems. This is extremely cost effective as the numerical cost of finding all the eigenvalues and eigenvectors of a real-symmetric $N \times N$ matrix is $O(N^3)$ Demmel (1997), so $N_1 = N/d$ implies the computational cost of the projection method is reduced by a factor of $d^3$.

*Author contributions.* All authors contributed to planning the work and writing and editing the manuscript.

*Competing interests.* The authors declare that they have no conflict of interest.

*Acknowledgements.* We gratefully acknowledge support from the Applied and Computational Analysis Program and the Arctic and Global Prediction Program at the US Office of Naval Research through grants N00014-18-1-2552, ONR Grant N00014-21-1-290, N00014-13-10291, N00014-15-1-2455, and N00014-18-1-2041. We are also grateful for support from the Division of Mathematical Sciences and the Division of Polar Programs at the US National Science Foundation (NSF) through Grants DMS-0940249, DMS-2136198, DMS-2111117, DMS-2206171, DMS-1715680, and DMS-1413454. Finally, we would like to thank the NSF Math Climate Research Network (MCRN), and especially Chris Jones, for supporting this work.





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
