# Peer review of "Stieltjes Functions and Spectral Analysis in the Physics of Sea Ice"

_Nonlinear Processes in Geophysics, 2022_

## Referee Comment (RC1)

**Review of "Stieltjes Functions and Spectral Analysis in the Physics of Sea Ice"**

This is a highly mathematical article about the exploitation of Stieltjes functions for the analysis of the physical properties of sea ice as an example of generic composite media. The high level of mathematics makes it difficult to follow some of the arguments. I therefore suggest that verbal explanations be attached to some of the more esoteric mathematical developments. For example, when the function F(s) is first introduced in Eq. (5), a simple explanation should be given as to how this function is obtained, as is done in Bergman (1978). Similar explanations should be attached to Theorem 1 and Theorem 2.

Many publications are cited in an attempt to outline the history of the topic of the current article. However, some citations are missing:

1. "Scattering electromagnetic eigenstates of a two-constituent composite and their exploitation for calculating a physical field", Bergman, Chen, and Farhi, Physical Review A 102, 063508 (2020).
2. "Eigenstates of Maxwell's equations in multi-constituent microstructures", Bergman, Physical Review A 105, 062213 (2022).

In these articles the eigenstates of Maxwell's equations are introduced for two-constituent and multi-constituent microstructures and used to calculate the local physical field.

In Line 94 Bergman (1982) should be cited.

In Lines 189 and 442, where elastic properties are mentioned, the following articles should be cited:

1. Y. Kantor and D. J. Bergman, Elastostatic resonances: a new approach to the calculation of the effective elastic constants of composites, J. Mech. and Phys. of Solids 30, 355-376 (1982).
2. Y. Kantor and D. J. Bergman, Improved Rigorous Bounds on the Effective Elastic Moduli of a Composite Material, J. Mech. and Phys. Of Solids 32, 41-62 (1984).

In these articles the Stieltjes method was first applied to elastic properties of microstructures.

---

## Referee Comment (RC2)

**Review of *Stieltjes Functions and Spectral Analysis in the Physics of Sea Ice by Kenneth M. Golden et al.**

February 24, 2023

**1    Overview**

This manuscript is a review paper on how various physical properties of sea ice, which is considered either as a microstructured composite or a uniaxial polycrystaline material. In either scenario, the analytic continuation method (ACM) has been applied successfully to 'parameterize' the various physical properties of the sea ice pertinent to the climate modeling in terms of the spectrum of the underlying self-adjoint operator defining the effective properties.

In this manuscript, some results obtained by applying this ACM approach for sea ice are shown by the authors to be counterparts in other fields such as the Anderson localization. These parallels are intellectually very interesting and beautiful. The preprint starts with the percolation models regarding the volume fraction and the electric conductivity of sea ice, followed by electromagnetic properties of sea ice as a two-phase composites of isotropic brine and ice, and then as uniaxial polycrystals (crystals with two different values among all possible crystal axes). In these sections, the focus is on how the microstructure is coded in the effective properties via the Stieltjes function in the ACM. In Section 5, the bounds derived from the ACM method are used to solve the inverse homogenization problems for using measurement of effective properties to bound the range of volume fractions. The last two sections, Sections 6 and 7, are devoted to the topic of ACM for diffusive processes and the Random matrix theory for sea ice physics, where the transition of the Eigenvalus Spacing Distribution (ESD) from Poissonian to universal Wigner-Dyson (WD) statistics of Gaussian Orthogonal Ensembles (GOE) for sea ice near the percolation threshold $\phi \approx 0.50$ is demonstrated by applying ACM to the CT image of sea ice. Also included are the project method describe in the appendix.

**2    Specific comments**

Topic-wise, this is a very good paper. All the results are from refereed papers. However, there are several points that need to be clarified.

1. In Figure 3, is the noise level known? Also, in the caption, it states "... a delta function singularity in the spectral function $\mu(\lambda)$ develops at $\lambda = 0$'. However, it is not clear to me how to see this in the graph of the spectral measure; indeed, it looks like the graph for $\phi = 0.12$ has the biggest jump near $\lambda = 0$ among all the three cases.

2. In several places in the manuscript, the word 'connectivity' appears several times. For example, in the paragraph between line 260 and line 264. It seems that as the volume fraction approaches 0.51, the gap vanishes and delta-measure starts to appear near the spectral end-point. But why is this a SIGNATURE of the composites being more connected? Again, in Line 395, it is stated that 'Connectivity information is also embedded in the spectral measure.' How do we measure the 'connectedness' from the spectral measure?

3. The entire Section 6 is very disconnected from the rest of the manuscript, partly because there are many un-defined symbols and missing essential details. For example, in Line 610, it states 'Non-dimensionalizing and homogenizing (26) shows...', but it is not clear what is the parameter over which the (26) is being homogenized; what is small? Or in Line 611, $\tilde{\phi}$ is mentioned but where to be found in the following equation, which is eqn (28). In Line 620, 'over the space-time period cell for periodic flows', why suddenly the average over space and TIME? Section 6 needs to be carefully rewritten to ensure the clarity and the consistency of the symbols and notations.

4. It would be great if the authors could add a paragraph or two elaborating on how these ACM results can directly contribute to the current climate model, and mention any known limitation, if any, of the ACM method in the context of sea ice modeling.

**3   Technical comments**

1. Line 148, please add references after 'its role in climate'.

2. Eqn. (21): $\frac{\partial}{\partial s}$ should be changed to $\frac{d}{ds}$.

3. Line 451: Please add references to the end of the sentence "These relations estimate permeability of a porous material characterizing ....'

4. Page 9, Figure 6: The numbering of these figures needs to be corrected.

5. Line 536: Is 'c-axis' crystal axis?

---

## Author Response (AR1)

**Response to the referees regarding the revised manuscript.**

First, now that we have gone through and revised the manuscript in detail in accordance with the referee comments, we believe the paper has been significantly improved and strengthened by addressing all of the excellent remarks. We thank both referees for their suggestions, and in a nutshell, we have taken all their suggestions and made all the requested changes in the revised manuscript. We have also added new, unpublished versions of several of the figures, and added many new references to that are relevant to the topics we consider.

Blue font is used in the manuscript to denote changes.

Response to Referee #1.

Thanks again for all your suggestions. We have included the requested references, added citations in the requested locations, and briefly described this type of spectral method. We have also now included intuitive descriptions of Theorems 1 and 2, and the function F(s) in Equation (5).

Response to Referee #2.

Thanks again for all your suggestions. Concerning points 1 and 2, the figure captions have been re-written to make these excellent issues more clear, particularly the caption in Figure 3. Also, concerning the excellent question as to why connectedness is signaled by the appearance of point measures near a spectral endpoint, from an intuitive standpoint it's helpful to keep in mind that F(s)=p/s is the exact expression for a geometry with layers parallel to the applied field, where p is the volume fraction of the conducting phase. This is the simplest composite geometry where the conducting phase connects across the sample, parallel to the field, and whose spectral measure is a single Dirac point measure of mass p at the spectral endpoint $\lambda$=0. Perturbing this situation, while maintaining the connection across the sample, keeps similar spectral behavior near the endpoint. This singular behavior there goes away, and is replaced by a spectral gap, as the connectivity of the conducting phase across the sample is destroyed. For point 3, we have substantially re-written the advection diffusion section to address the points raised, and for point 4 we have added new material in the Introduction to address the issues brought up there, which we believe substantially improves the readability and applicability of the paper. All the technical comments were taken care of as requested. Thank you very much!

---

## Author Response (AR2)

We thank the editor for the suggestion to shorten sentences, clarify, etc. We have gone through the paper in detail and done this to make it we hope much more readable. Thanks!!